# Bovine milk consumption affects the transcriptome of porcine adipose stem cells: Do exosomes play any role?

Katherine Swanson[1], Jimmy Bell[2], David Hendrix[2], Duo Jiang[3], Michelle Kutzler[1], Brandon Batty[1], Melanie Hanlon[4], Massimo Bionaz [1] *

**1** Animal and Rangeland Sciences, Oregon State University, Corvallis, Oregon, United States of America, **2** Biochemistry and Biophysics, Oregon State University, Corvallis, Oregon, United States of America, **3** Statistics, Oregon State University, Corvallis, Oregon, United States of America, **4** Food Science and Technology, Oregon State University, Corvallis, Oregon, United States of America

\* massimo.bionaz@oregonstate.edu

**Data Availability Statement:** RNAseq files are available from the Sequence Read Archive (NCBI) database (accession number(s) PRJNA1083710).

## Abstract

The potential association of milk with childhood obesity has been widely debated and researched. Milk is known to contain many bioactive compounds as well as bovine exosomes rich in micro-RNA (miR) that can have effects on various cells, including stem cells. Among them, adipose stem cells (ASC) are particularly interesting due to their role in adipose tissue growth and, thus, obesity. The objective of this study was to evaluate the effect of milk consumption on miR present in circulating exosomes and the transcriptome of ASC in piglets. Piglets were supplemented for 11 weeks with 750 mL of whole milk (n = 6; M) or an isocaloric maltodextrin solution (n = 6; C). After euthanasia, ASC were isolated, quantified, and characterized. RNA was extracted from passage 1 ASC and sequenced. Exosomes were isolated and quantified from the milk and plasma of the pigs at 6–8 hours after milk consumption, and miRs were isolated from exosomes and sequenced. The transfer of exosomes from milk to porcine plasma was assessed by measuring bovine milk-specific miRs and mRNA in exosomes isolated from the plasma of 3 piglets during the first 6h after milk consumption. We observed a higher proportion of exosomes in the 80 nM diameter, enriched in milk, in M vs. C pigs. Over 500 genes were differentially expressed (DEG) in ASC isolated from M vs. C pigs. Bioinformatic analysis of DEG indicated an inhibition of the immune, neuronal, and endocrine systems and insulin-related pathways in ASC of milk-fed pigs compared with maltodextrin-fed pigs. Of the 900 identified miRs in porcine plasma exosomes, only 3 miRs were differentially abundant between the two groups and could target genes associated with neuronal functions. We could not detect exosomal miRs or mRNA transfer from milk to porcine-circulating plasma exosomes. Our data highlights the significant nutrigenomic role of milk consumption on ASC, a finding that does not appear to be attributed to miRs in bovine milk exosomes. The downregulation of insulin resistance and inflammatory-related pathways in the ASC of milk-fed pigs should be further explored in relation to milk and human health. In conclusion, the bioinformatic analyses and the absence of bovine exosomal miRs in porcine plasma suggest that miRs are not vertically transferred from milk exosomes.

**Funding:** USDA National Institute of Food and Agriculture (Washington, DC) Agriculture and Food Research Initiative Exploratory, grant # 2015-67030-23872, along with the National Needs Graduate Fellowships, Grant #2014-38420-21800 The funders had no role in study design, data collection, and analysis, the decision to publish, or preparation of the manuscript.

**Competing interests:** The authors have declared that no competing interests exist

## Introduction

Obesity, especially childhood obesity, is a serious concern due to its links with cardiovascular disease, insulin resistance, and type II diabetes [1]. Adipose-derived stem cells (**ASC**) are mesenchymal stem cells that can differentiate into preadipocytes, osteocytes, chondrocytes, or myocytes [2]. Metabolic status and a difference in gene and hormone expression can determine which cell lineage these stem cells will commit to [2]. For preadipocytes, a metabolic environment characterized by a significant time of increased energy intake and increased glucose uptake appears to stimulate ASC differentiation [3]. This increase in adipogenic differentiation (or adipogenesis), along with the accumulation of lipid droplets in mature adipocytes (or lipogenesis), leads to obesity [3, 4].

While the cause of obesity can be related to various factors such as genetics, epigenetics, and environment, one of the most critical factors is diet [5, 6]. Nutritional factors such as protein and fat drive adiposity [7, 8]. Those nutritional factors are abundant in milk, the first food humans consume, and many humans continue to consume milk throughout their lives. It has been argued that proteins in milk can increase the activation of the Mammalian Target of Rapamycin Complex 1 (mTORC1) pathway that, in turn, can upregulate lipogenesis [7] in adipocytes, leading to an increase in obesity. This does not appear to be supported by clinical studies where data indicated that drinking whole cow's milk at a young age reduced the risk of children developing obesity [8, 9]. Due to the limitations associated with clinical studies, conclusive proof of whether or not milk directly affects adipogenesis and lipogenesis, and in turn obesity, is lacking and, if so, the mechanism behind this effect.

In addition to the concern regarding milk and obesity, there has been more focus on the role sugary beverages might play. Researchers have found an association between increased consumption of sugary beverages and obesity [10]. Similar studies have been conducted to assess the correlation between milk consumption and sugary beverages on obesity in children [11, 12]. These studies found that consuming milk was correlated to an increase in lean body mass in contrast to the correlation between consumption of sugary beverages and an increase in obesity [11, 12]. None of these studies could directly prove the effect of milk or sugary drinks on the incidence of obesity.

Milk is a complex nutritional product containing many bioactive molecules [13]. One of the bioactive molecules discovered in milk are micro RNAs (miRs) in exosomes. miRs are small RNAs that suppress the expression of specific target genes through post-transcriptional silencing [14]. Exosomes are microvesicles that are around 100 nm in diameter that work as cargo vesicles for several oligonucleotide molecules between cells, including miRs [15]. Various studies have provided evidence that these exosomal miRs are stable under degradative conditions and can be taken up by the cells of non-origin species *in vitro* [16, 17] and can be absorbed from milk in the gastrointestinal tract (i.e., vertically transferred) [17, 18]. Milk exosomes are enriched with miRs that target genes associated with the immune response [19]; however, bovine milk exosomes also contain miRs that can target genes related to adipogenesis, such as miR200c or miR-21 [20, 21]. Therefore, miRs in bovine milk exosomes may be transferred to humans consuming that milk, affecting adipogenesis.

To our knowledge, there are no *in vivo* studies investigating the effect of bovine milk on ASC. primarily via an impact on the transcriptome by changes in plasma miRs profile after consuming the milk. Given the presence of adipogenesis-related miRs in bovine milk, our study aimed to explore the possibility of these miRs being vertically transferred from plasma to circulation and, ultimately, ASC in the animals consuming milk. Using growing piglets as our model for young children [22], we sought to determine whether the oral consumption of bovine milk compared to an isocaloric maltodextrin supplementation could affect the quantity

of ASC in adipose tissue and their adipogenesis/lipogenesis capacity and transcriptome. In addition, we aimed to determine if consumption of bovine milk affects the profile of miRs present in plasma exosomes that might, in turn, affect ASC adipogenesis/lipogenesis. We hypothesized that milk consumption compared to isocaloric maltodextrin supplementation affects the adipogenic capacity of ASC via changing the miRs profile of circulating exosomes that, in turn, affect ASC transcriptome.

## Methods

### Animals and experimental design

Experimental procedures used in this study were approved by the Institutional Animal Care and Use Committees of Oregon State University (protocol #4691). The *in vivo* experiment was conducted between January and March of 2017.

Details of the animals used, and experimental design were previously described (see Study 2 in [23]). Twelve Duroc-Berkshire male piglets 8–9 weeks of age were randomly assigned to groups blocking for weight and litter for an 11-week trial. The piglets were kept in the Hogg Animal Metabolism Barn at Oregon State University in 1.3 m$^2$ pens with the ground lined with AstroTurf. The room had an air conditioner system, and the temperature was maintained at 20˚C. All pigs had unlimited access to water and were fed every morning, as previously described [23]. In addition to the feed, the piglets received an additional isocaloric supplement of either a maltodextrin solution (**C**; cat# 007-345-0341, Honeyville, USA) or whole fresh raw cow milk collected daily from the Oregon State University Dairy bulk tank (**M**). The milk samples collected over three days during the experiment were used for exosome isolation (see below). The maltodextrin solution was made every 3–4 days and kept refrigerated when not being fed. Milk was fed at 750 mL/d (4.8% fat, 3.6% protein, 4.8% lactose) while maltodextrin solution at 500 mL/d. The maltodextrin solution amount was calculated based on the number of calories in the milk (576.3 kcal in 750 mL), given that maltodextrin has an energy density of 4 kcal/g. Each piglet received 144 g of maltodextrin mixed in 500 mL of water to make it a palatable consistency. The supplements were fed in small troughs separate from their feed troughs, and the piglets were given their food after they drank their supplement, usually within 1–2 minutes.

On the last day of the study, blood was drawn from the jugular of each piglet using 22G 3.5" hypodermic needles (cat# N22312, Air-Tite Vet, Virginia Beach, VA) into 10 mL heparin tubes at 6 hours after the supplements were fed. The heparin tubes were centrifuged at 3000×g for 10 minutes to separate the plasma for exosome and miRs isolation. At the end of the 11 weeks, the piglets were euthanized via a jugular injection of euthasol (Somnasol Euthanasia, cat# EU-HS-045-100-0, HenrySchein, USA).

### Exosome isolation from porcine plasma and bovine milk

For both the pig serum and bovine milk, exoRNeasy Serum/Plasma Midi Kits (cat# 77044) or exoEasy Maxi Kits (cat# 76064) from Qiagen (Germany) were used to isolate exosomes and extract miRs from the exosomes, respectively. To ensure the milk was free of milk fat globules, cells, and cellular debris, it was first poured into a 50 mL conical tube (cat# 89039–656, VWR, USA) and centrifuged at 3000×g for 10 minutes at room temperature to allow the top cream layer to be removed with a metal spatula. Then, it was poured into a new 50 mL conical tube and centrifuged again at 3000×g for 10 minutes to remove the rest of the cream. Then, 5 mL of the supernatant was transferred into a new ultracentrifuge tube. These tubes were centrifuged at 12000×g for 1 hour in a JA-20.1 fixed angle rotor in a Beckman Coulter Optima L-100K ultracentrifuge. The tubes were transferred to a 50 TI rotor and centrifuged for 1 hour at

35000×g. Using the same rotor, the tubes were centrifuged for 1 hour at 70000×g. Then, the clear supernatant was transferred to a new sterile 15 mL tube (cat# 89039–664, VWR, Radnor, PA) for exosome and miR isolation. After ultracentrifugation, the exosomes from the bovine milk were isolated using the same protocol to isolate exosomes from the blood plasma. Samples were filtered using a 0.8 μm syringe filter (Millipore Millex-AA SLAA033SB) into a new 15 mL tube. The isolation of exosomes was performed following the manufacturer's instructions. For the samples used for only exosome isolation, 1mL of buffer XE was added to the spin column and centrifuged at 2000×g for 5 minutes to elute the exosomes. The flow through was reapplied to the spin column and centrifuged at 4300×g for 5 minutes. The eluted exosomes were stored at -20˚C for further analysis. Exosomes were quantified, and particle size distribution was analyzed using ZetaView (Particle Matrix, Germany).

To isolate miRs, the spin column containing purified exosomes was transferred to a new collection tube, and 700 μl of QIAzol was added to the column before it was centrifuged at 4,300×g for 5 minutes. The isolation of RNA was performed following the manufacturer's protocol. The samples in RNase-free water were then stored at -80˚C until further analysis.

## Exosome marker confirmation

Isolated exomes were confirmed using the Exo-Check antibody array kit (cat# EXORAY200B-4, Cat #EXORAY210B-8) from System Biosciences (Palo Alto, CA) that checks for exosome markers CD63, CD81, ALIX, FLOT1, ICAM1, EpCam, ANXA5 and TSG101. Previously isolated exosomes stored at -80˚C were used for this assay. The exosomes were stored in 1X PBS and confirmation of exosome markers was performed following the manufacturer's protocol. The developed membrane was imaged using luminescence capture in a MultiImage II, Alpha Innotech.

## Adipose stem cell isolation

Once the piglets were euthanized, subcutaneous back fat from each piglet was dissected using a sterile procedure. The corps of the animals were hung with their heads down and bled by severing both jugular veins. The area encompassing the back, loin, and rump was clipped, followed by sterile gauze immersed in Povidone scrub (cat#3659, HenrySchein, USA) to clean the area. Then, a sterile gauze immersed in 70% ethanol was used to complete the cleaning of the area. The procedure was repeated once, and 70% ethanol was sprayed on the whole area that was to be collected. A butcher knife was cleaned using Povidone scrub followed by 70% ethanol, and the entire area, including the skin and the subcutaneous fat, was dissected. The tissue was then "closed as a book," ensuring the sterile backfat was internal and the skin was external. The whole dissected tissue was put in a large, fresh Ziploc bag and stored on ice for 4 hours before beginning the stem cell isolation. The ASC were isolated as previously described [24]. Briefly, a sterile layer of fat was separated and weighed in pre-weighed sterile petri dishes (cat# 32-107G, Genesee Scientific, USA) before being minced using a sterile scalpel blade and digested for 45 min in a shaker in 0.075% collagenase (Collagenase Type I, Clostridium histolyticum; cat# J62406, Alfa Aesar, USA) at 37˚C. The resulting cells were then passed through a 70 μM cell strained (cat# 10199–656, VWR, USA), red blood cells lysed using a red lysis buffer (cat# 89131–000, VWR, USA), and cells collected in 20 ml DMEM (cat# 25–500, Dulbecco's Modified Eagle's Medium, Gen Clone, USA) containing 10% FBS (cat# S11150, Atlanta Biologicals, USA), 10000 U/mL of PennStrep (cat# 25–512, Gen Clone, USA), and 3% amphotericin (cat# ABL01-100ML, Caisson, USA). The tubes containing the cells in DMEM were centrifuged at 400×g for 5 minutes, followed by resuspension of the cell pellet in 10 mL of DMEM.

To count viable nucleated cells, 75 μL of isolated cells in media was added to 75 μL of a staining solution containing one drop of NucBlue™ Live ReadyProbes™ Reagent (cat#R37605, ThermoFisher Scientific, USA) and one drop of NucRed™ Dead 647 ReadyProbes™ Reagent (cat#R37113, ThermoFisher Scientific, USA) in 500 μL PBS. Cells were incubated in the staining solution for 15 minutes before loading the cells into both counting chambers of a hemocytometer. Pictures for each of the 8eightcounting quadrant areas of the hematocytometer were taken using the DAPI and the Y55 (far red) filters in a robotic fluorescence microscope (Leica DMI6000B, Leica Microsystems, USA). The number of live cells were counted using CellProfiler [25]. Cells were then plated at 10000 viable cells per $cm^2$ in two 75 $cm^2$ flasks (cat# 10062–860, VWR, USA). The cells were placed in a humidified incubator and maintained at 39˚C with 5% $CO_2$, and the first media was changed after 24 hours. The remaining cells (i.e., passage 0) were immersed in 1 mL freezing media (cat# SH30894.01, HyClone™ AdvanceSTEM Cryopreservation Medium, GE Healthcare, USA), left at -80˚C overnight, and stored in liquid N.

## Colony forming unit (CFU) assay

The CFU was performed as previously described [24]. Briefly, harvested ASC were plated at a density of 25000 cells/25 $cm^2$ culture flask (cat# 82051-074, VWR, USA) with three technical replicates. After plating, the cells were incubated at 39˚C and 5% $CO_2$, with the first media change 24h later and subsequent media changes occurring every 72 hours. After 14 days of culture, colonies were fixed using 10% buffered formalin (cat# CA71007-348, VWR, USA) for 30 minutes, exposed to hematoxylin staining solution (cat# 3536–32, RICCA Chemical Company, USA) for 30 min, and washed with tap water. The stained colonies were then manually counted.

## Proliferation assay

To assess the growth of the ASC, the Vybrant MTT Cell Proliferation Assay was used (cat# V13154, Molecular Probes Inc). To ensure consistent density of cells between groups plated for this assay, passage one cells were used. The primary ASC were passed as previously described [24]. Briefly, the passage 0 ASC were harvested using 0.25% trypsin (cat# 02-0154-0100, VWR, USA) before being centrifuged at 1000×g and re-suspended in DMEM. The cells were then counted using a hematocytometer and plated at a 10000 cells/well density in two 96-well flat bottom plates with three technical replicates per animal. After 24 hours, one plate was used to count the number of attached cells using NucBlue™ Live ReadyProbes™ Reagent to account for any discrepancies in original number of cells plated. After 72 hours, the second plate was used with the Vybrant MTT Cell Proliferation Assay kit following the manufacturer's protocol. Briefly, MTT stock solution was added to the cells in fresh media, which were then incubated at 39˚C for 4 hours to label the cells. Then, DMSO was added to solubilize the formazan before reading the absorbance at 540 nm. Previously isolated porcine ASC plated at different known concentrations were used to make a standard curve to calculate the final amount of ASC/well. These values were then divided by the amounts counted at 24 hours and multiplied by 100 to give the percentage of proliferation over 48 hours.

## In vitro adipogenic and osteogenic differentiation

For differentiation, passage 1 ASC were plated in 48 well plates at a density of 40000 cells/well. Four plates were used for each differentiation assay to allow for the separate analysis of four time points: 0, 7, 14, and 21 days. The ASC from each pig were plated simultaneously, and the differentiation media was added at 80% confluence for the osteogenic assay and 90% confluence for the adipogenic assay. Osteogenic and adipogenic media were prepared as previously

described [24] with the adipogenic media containing one µM dexamethasone (cat#50-02-2, Alfa Aesar, USA), 0.5 mM 3-isobutyl-1-methylxanthine (cat# 28822-58-4, Acros Organics, USA), 200 µM indomethacin (cat# 53-86-1, Alfa Aesar, USA), and 0.5 µg/mL porcine insulin (cat # 12584-58-6, Sigma-Aldrich, USA). The osteogenic media was prepared to contain 0.1 µM dexamethasone, 10 mM ß-glycerophosphate (cat# A2253-0100, AppliChem, USA), and 50 µM ascorbic acid (cat# 50-81-7, G-Biosciences, USA). At each time point, cells from both assays were fixed by rinsing with DPBS before adding 10% buffered formalin for 30 minutes; at this point, the formalin was removed.

For adipogenic cells, the fluorescent Nile Red stain was used (cat# 22190, AAT Bioquest Inc., USA). Briefly, a 500 nM working solution was made by adding 2.38 µL of 10.5 mM stock solution to 50 mL DMEM. In addition, 100 drops of NucBlue nuclear stain were added to 50 mL DMEM to stain the nuclei of the cells. This working solution was then added to the fixed cells. The plates were incubated at room temperature for 12 min before removing the working solution and adding DPBS. The wells were then viewed using a robotic fluorescence microscope (Leica DMI6000B) to take nine pictures of each well using both a DAPI (blue) and N2.1 (red) filter to view both the nuclei and lipid droplets of each cell. These pictures were then analyzed using CellProfiler 3.0.0 to count the number of adipocytes, total cells, and lipid droplets/cell. The CellProfiler pipeline is available in **S1 File**.

For osteogenic cells, the Alizarin Red S stain (cat# 130-22-3, Amresco, USA) was used following a previously described protocol [24]. Osteogenic cells were viewed using the bright field setting of the microscope (Leica DMI6000B). The osteogenic assay was performed only to confirm the presence of typical osteogenic nodule formation [24] by mesenchymal stem cells. For this reason, osteogenesis was not quantified.

## RNA isolation

For the isolation of RNA from ASC, passage 1 cells were plated at 10000 cells/well, grown to 80% confluence (8–14 days), and harvested using 0.25% trypsin before being homogenized and stored in 1 mL TRIzol (cat# 15596026, ThermoFisher) in microcentrifuge tubes (cat# 22–281, Genesee Scientific, USA) at -80˚C. Total RNA was isolated using a previously described method [26] and cleaned using RNeasy Mini Kit (cat# 74104, QIAGEN, Germany) following the manufacturer's instruction (without DNA digestion). The RNA was eluted using 40 µl of RNase free water and stored at -80˚C until further analysis.

## RNA sequencing and analysis

The RNA samples were sent to the Center for Genomic Research and Biocomputing (now Center for Quantitative Life Sciences) at Oregon State University to be sequenced. The samples were run on the Illumina HiSeq 3000 as 150 bp paired-end runs (Illumina, USA). The RNAseq files were then aligned and annotated using Kallisto [27]. A Kallisto index file was created using a known transcriptome for *Sus scrofa* from NCBI (GCF_000003025.6_Sscrofa11.1_rna.fna, NCBI). Then, each pair of samples paired-end reads files were pseudo-aligned to the index file using default options in Kallisto, and 100 bootstraps were used for each run. The bootstrapping allows for repeat analysis with replacement to help account for technical variance and get rough confidence intervals. The resulting annotated RNA count estimates were used for the statistical analysis.

## miRs sequencing and analysis

The porcine plasma exosome miRs and bovine milk exosome miRs were sent to the Center for Genomic Research and Biocomputing (now Center for Quantitative Life Sciences) at Oregon

State University to be sequenced. The samples were run on the Illumina MiSeq 3000 as small RNA reads. The adapter sequences from the resulting read files were trimmed using cutadapt [28] with options set to trim each sequence at locations with less than a 20% mismatch from the original adapter sequence and to trim low-quality ends that did not meet a minimum quality score cutoff of 10. Trimmed read sequences were mapped to locations within their respective genomes using bowtie [29] with a seed length 22 and returning alignments with 2 or fewer mismatches. Reads were aligned to the Sscrofa11.1 genome for pig samples and the UMD3.1.1 genome for cow milk samples. After the reads were mapped, they were annotated using miR-Base v21 [30]. Homologous and novel miRs were found using miRWoods [31]. The sequence alignments used in miRWoods predictions were produced according to the miRWoods manual and had slightly different settings for cutadapt and bowtie. Trimmed reads from cutadapt were suppressed if they were less than 17 nucleotides in length, and a script from the miRwoods package was run to remove trimmed reads with average quality scores of less than 30. Bowtie was run with settings that searched for alignment within a seed length of 18, a maximum mismatch quality score of 50, and only reported alignments from the best stratum of alignments with 10 or fewer mappings to the genome.

## Milk miRs transfer to porcine plasma

Three 5-week-old Yorkshire Hampshire piglets (11.1±0.19 kg) were enrolled, and whole fresh milk from the bulk tank of the Oregon State University Dairy Farm was used. The piglets were kept in a single pen during the 4-day adaptation period, where an increased amount of milk was provided in a single plastic feeder (**S1 Fig** in **S2 File**). After the piglets could completely drink 1.2 L in <20 min as a group, they were separated into single pens. There a second 2-day adaptation period was performed before starting the trial. On the first day, 400 mL and 500 mL of milk were provided at 9:00 AM on the second day. The time for the complete consumption of milk was recorded. All animals could consume ≥400 mL of milk in <6 min during the second day. For this reason, the study was started after six days of adaptation by providing 700 mL of milk to each animal at 9:00 AM. Blood was collected as described above at time 0 (i.e., just before giving the milk) and then at 1, 2, 4, and 6 hours after providing the milk. The time of blood collection was based on a similar prior study performed in humans [17]. Milk that still needed to be consumed was measured at each blood collection to determine the rate of milk consumption. Pig chow was provided to the piglets once most of the milk was consumed. As described above, the milk exosomes and the RNA in the exosomes were isolated from a sample of the milk fed to the piglets and from the plasma samples. Primers to amplify specific miRs were designed and ordered via QIAGEN. cDNA from the isolated RNA was performed using miScript II RT kit (cat# 218160, QIAGEN) following the manufacturer's instruction. Briefly, for a 20 μL total reaction per sample, a total of 60 ng of RNA was used with 4 μL of the 5x miScript HiSpec buffer, 2 μL of 10x miScript Nucleics Mix, 2 μL of miScript Reverse Transcriptase mix, and brought to 20 μL volume using RNase/DNase free water. Following the manufacturer's guidelines, the PCR reaction was performed using the miScript SYBR Green PCR Kit (cat#218073, QIAGEN). Briefly, a 10 μL reaction was performed in triplicate with 3 μL cDNA (diluted 4-fold using RNase/DNase free water), 5 μL SYBR green, 1 μL of both 10x miScript Universal Primer and 10x miScript Primer. The primers to amplify *Sus scrofa* Let7b, miR1285, miR148A, miR200a, miR200c, miR22, and miR29b were designed and produced via QIAGEN custom miScript Primer Assay. Two bovine transcripts (*CSN3* and *LALBA*) plus porcine *GAPDH* were also measured in the plasma exosomes from pigs. RTqPCR methods and primers were as previously described [26, 32]. Both PCR assays were performed using an ABI 7900HT system with 40 cycles, and the final data was obtained using LinRegPCR [33].

## Statistical analysis

All data, except RNA sequencing data, was analyzed using SAS (v9.4, SAS Institute, USA). Data was checked for outliers using Prog Reg. Data with studentized $t \geq 3$ were removed for subsequent statistical analysis. Datasets without a time component were analyzed using a Proc GLM with treatment as the main effect and pig as the random effect. For the exosome size, GLM analysis was performed in each separate bin. For adipogenesis-related data, a GLIMMIX model was used with a heterogenous autoregressive covariate model (chosen after checking for the smaller Akaike Information Criterion and the Generalized Chi-Square/degree of freedom ratio close to 1). A p-value of $\leq 0.05$ denotes a biologically significant effect, while p-values $>0.05$ and $\leq 0.1$ denote a trend.

A negative binomial model was fitted to the data using the edgeR package in R for mRNA and miR transcriptomes. The resulting output included transcripts differentially expressed (DEG) amongst the treatment group compared to the control group, associated p-values, and false discovery rate (FDR).

## Functional analysis of differentially expressed RNA

For mRNA transcriptome, only transcripts with an FDR < 0.1 were further analyzed using the Dynamic Impact Approach (DIA) [33] and the Database for Annotation, Visualization, and Integrated Discovery (DAVID) v6.8 [34]. For DIA, the *Sus scrofa* gene IDs were converted to *Homo sapiens* gene IDs by using the biological DataBase network (https://biodbnet-abcc.ncifcrf.gov/) [35]. The annotated transcriptome (i.e., all transcripts detected in ASC) was used as the background for DIA and DAVID. Results from DAVID were downloaded using the "cluster" and "chart" options.

For miRs with a significant difference in abundance in plasma exosomes between groups, downstream target genes were predicted using TargetScanHuman, v. 7.2 [36]. Functional enrichment analysis of the target genes was performed by DAVID using the whole human transcriptome as background, considering all predicted targets and, to minimize false positives, also predicted target with a cumulative weighted context++ score of $\leq$-0.4. Enrichment of miRs in the DEG in ASC isolated from piglets supplemented with milk vs. the ASC isolated from piglets supplemented with maltodextrin was performed using MIENTURNET [37].

## Effects of milk-specific miRs on ASC using gene reporters

To assess the activity of milk-specific miRs on ASC as previously performed [17], reporter gene plasmid for miR-29b and miR-200c were generously provided by Dr. Janos Zempleni (University of Nebraska–Lincoln). However, due to the known poor transfection efficiency of primary cells, especially mesenchymal stem cells [38], a preliminary study was performed to assess the efficiency of plasmid transfection using a GFP-expressing plasmid and a series of commercially available transfection reagents (see **S1 Table** and **S2 Fig** for details of methods, results, and discussion in **S2 File**). None of the transfection reagents tested provided >2.5% transfection efficiency, which is the minimum required to obtain reliable data with luciferase assays [39]; thus, we did not proceed with the miR-reported gene assay.

# Results

## Porcine ASC abundance and capacity of differentiation toward the adipogenic lineage

ASC isolated from pigs fed milk tended (P = 0.10) to have a higher amount of ASC isolated per gram of tissue and a higher amount of CFU and ASC proliferation than control pigs (**Fig 1A–1C**). Isolated ASC successfully differentiated into adipocytes and osteocytes (**S3, S4 Figs** in

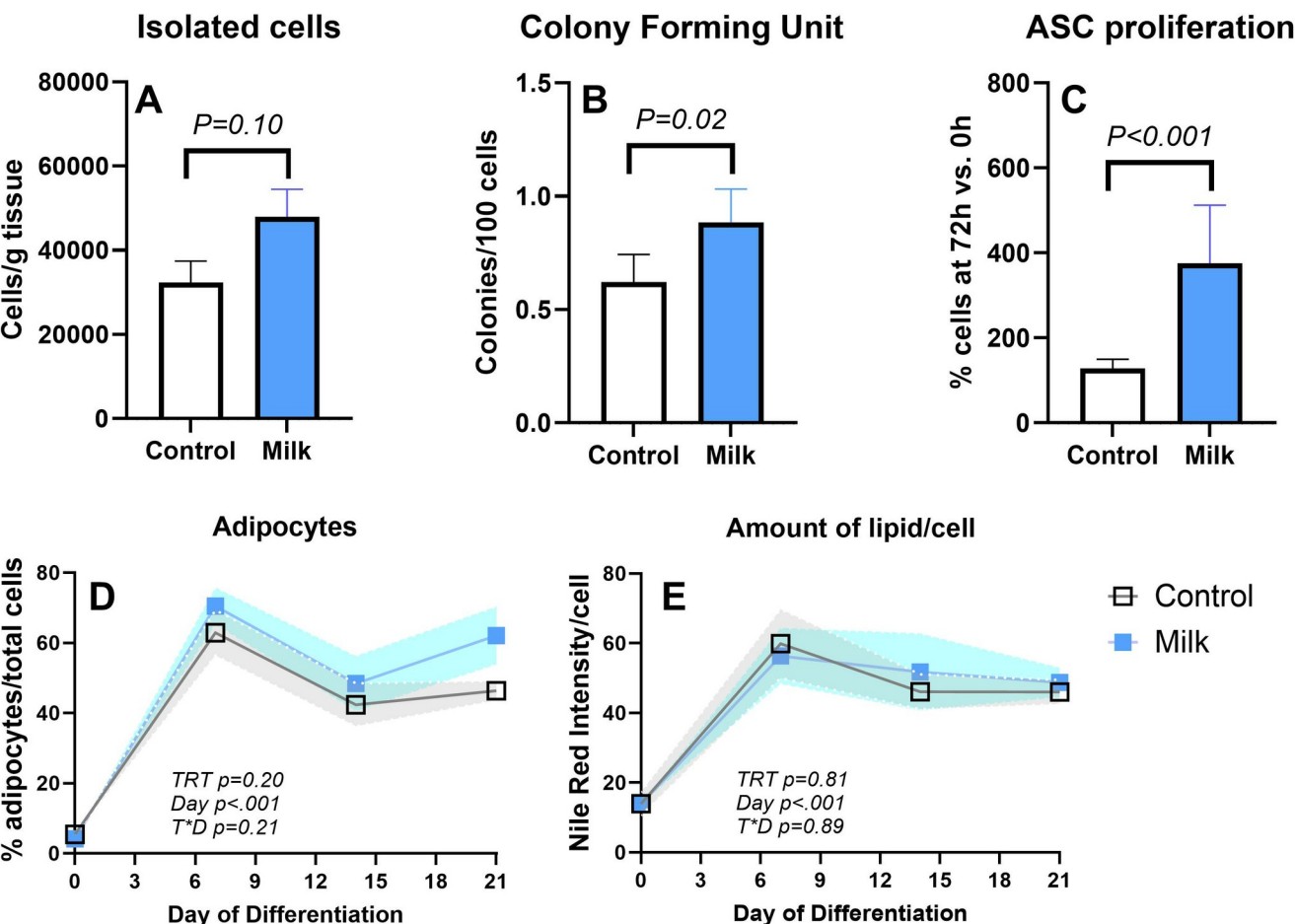

**Fig 1. Effect of milk consumption on adipose stem cells.** (A) number of isolated adipose stem cells (ASC); (B) colony forming unit of ASC; (C) proliferation of ASC; (D) differentiation; and (E) amount of lipid/ASC. ASC were isolated from 6 piglets supplemented with milk and six control piglets supplemented with maltodextrin.

S2 File). The latter was observed by the formation of the typical large osteogenic nodules [24]. After adding the adipogenic media, the number of cells presenting lipid droplets (i.e., adipogenesis) significantly increased, but no differences were observed for the proportion of formed adipocytes or lipid accumulation/cells (**Fig 1D and 1E**).

### Exosome isolation from porcine plasma and bovine milk

Exosomes isolated from porcine plasma and bovine milk were positive for all measured exosome markers (**S5 Fig** in **S2 File**). Furthermore, the transcription of positive and negative markers of mesenchymal cells [40], especially CD90, and CD29, which have been used previously to isolate porcine ASC [41] revealed that the isolated cells have a high expression of CD29, CD90, and STRO-1 and low or no expression of negative markers (**S6 Fig** in **S2 File**), supporting positive isolation of ASC. The measurement of exosomes using ZetaView revealed a larger concentration of exosomes in porcine plasma compared to whole bovine milk but only a numerically larger concentration of exosomes isolated from plasma collected from pigs fed with milk compared to plasma from the control pigs (**Fig 2A**).

For the exosome size distribution, the bovine milk had a higher frequency of exosomes in the 40 and 90 nm bins but a lower concentration of exosomes with >90 nm in diameter when

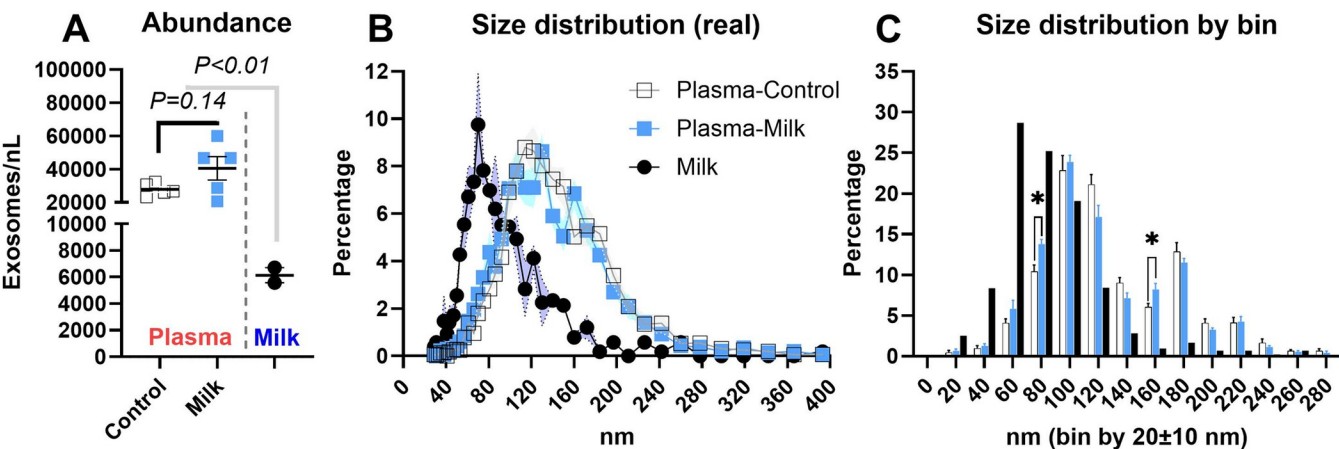

**Fig 2. Exosomes size distribution in bovine milk and pig plasma.** (A) Abundance, (B) size distribution as diameter shown as continuous, and (C) distribution of diameter shown as incremental (20±10 nm) range of sizes (i.e., bins) of exosomes isolated from the cow's milk (n = 2) provided to the piglets and from the plasma of the piglets 6–8 hours after consuming either milk (n = 5) or maltodextrin (n = 5; control). *indicates P<0.05 between the percentage of exosomes in the plasma of control- or milk-fed animals.

compared with plasma (**Fig 2B**). In addition, the exosomes isolated from the plasma of pigs fed milk had a higher frequency of exosomes in the 80 and 160 nm bins with a tendency for a lower frequency of exosomes in the 120 (P = 0.06) and 140 (P = 0.07) nm bins when compared to exosomes isolated from the plasma of control pigs (**Fig 2C**).

## Porcine ASC mRNA expression

The RNA integrity number (RIN) was 8.9±0.7. There were 508 differentially expressed genes (DEG) with FDR≤0.1 affected by milk supplementation (**S3 File** and **S7 Fig** in **S2 File**). Overall, 236 DEG were down-regulated, and 272 DEG were up-regulated in ASC isolated from pigs supplemented with milk vs. ASC isolated from pigs supplemented with maltodextrin (control).

**Dynamic Impact Approach analysis of DEG.** The complete results of the DIA analysis are available in **S4 File**. The summary of KEGG pathways affected by the DEG as analyzed by DIA is reported in **Fig 3**. The most impacted category of pathways by the DEG between ASC isolated from pig supplemented with milk vs. maltodextrin (control) was the 'Organismal Systems,' which was overall inhibited by feeding milk, followed by 'Metabolism' (activated) and 'Environmental Information Processing' (inhibited). Within the 'Organismal Systems' category, many immune-related pathways were inhibited for pigs fed milk vs control. These included pathways related to T cell differentiation and signaling, B cell signaling, and some antigen processing and phagocytosis pathways. The most notable pathways activated within the 'Metabolism' category for pigs supplemented with milk included those related to ether lipid metabolism, propanoate metabolism, and glycerophospholipid metabolism. Within the 'Environmental Information Processing' category, the VEGF signaling pathway was inhibited in pigs supplemented with milk while the AMPK signaling pathway was activated.

The 40 most affected pathways with related categories and sub-categories of pathways (except several sub-categories that are not relevant for ASC, such as digestion and absorption-related pathways, porphyrin and chlorophyll metabolism, addictive substances, and several human diseases) are reported in **Fig 4**. DIA analysis of the effect of DEG on specific pathways indicated that milk supplementation induced pathways associated with accumulation of lipids (i.e., glycerophospholipid metabolism) and inhibited synthesis of some of the glycans.

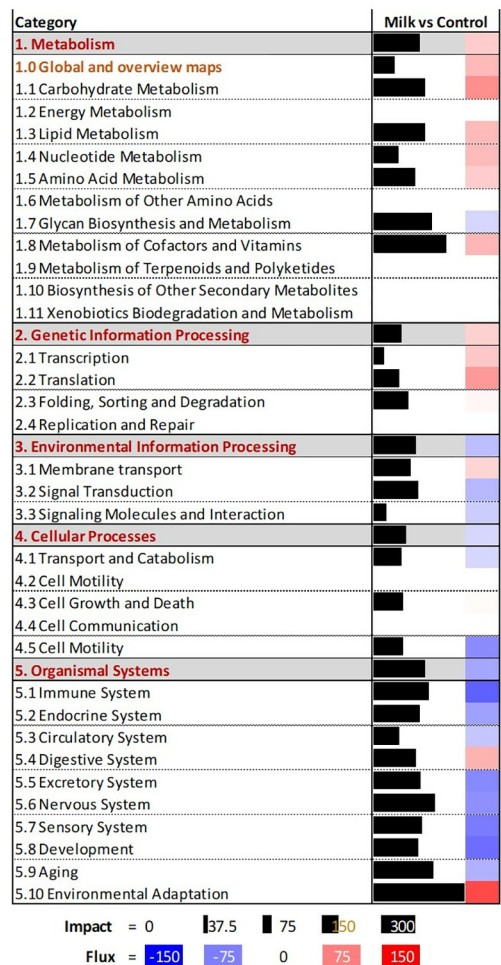

**Fig 3. Summary of categories and sub-categories of KEGG pathways from the analysis of differentially expressed genes in adipose stem cells.** Adipose stem cells were isolated from piglets consuming milk (n = 6) or piglets consuming maltodextrin (n = 6; control). The analysis was performed using the Dynamic Impact Approach. Shown are the impact (black horizontal bar; the larger the bar, the larger the impact) and the flux (or direction of the impact; red denotes activation and blue inhibition) of the overall categories of pathways.

Pathways inhibited by feeding milk were those associated with signaling, particularly related to VEGF, ErbB, phosphatidylinositol, and MAPK. Milk consumption activated pathways related to hedgehog, notch, and AMPK. Pathways that were inhibited in ASC by feeding milk were most of the immune-related pathways, such as Th cell differentiation, T cell receptor signaling, and antigen processing and presentation. In addition, pathways associated with insulin resistance and type II diabetes were inhibited in ASC from pigs supplemented with milk compared to ASC from the control pigs.

**DAVID analysis of DEG.** The complete results of the DAVID analysis are available in **S5 File**. A larger number of terms were enriched in down-regulated DEG than up-regulated DEG. In **Fig 5** are reported the Gene Ontology Biological Process terms (GO) and KEGG pathways that were enriched with an EASE score<0.05 in both up- and down-regulated DEG. Up-regulated DEG enriched a few GO terms related to several categories; however, of note is the significant enrichment of DEG associated with the negative regulation of insulin signaling. In

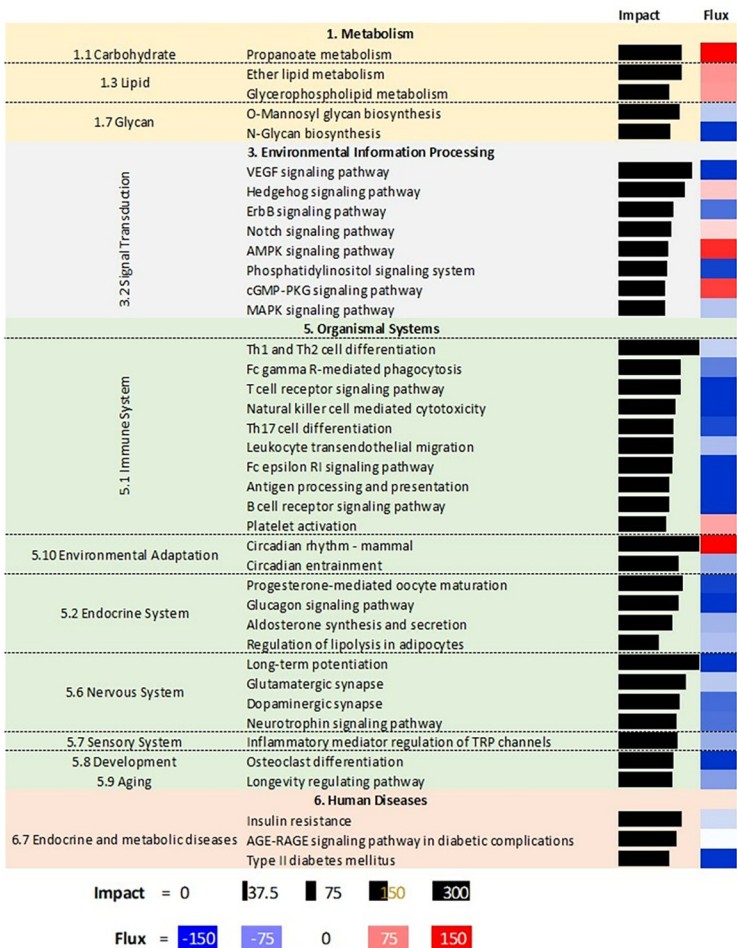

**Fig 4. Most impacted KEGG pathways in adipose stem cells by feeding milk to piglets.** The analysis was performed using the Dynamic Impact Approach. Shown are the impact (black horizontal bar; the larger the bar, the larger the impact) and the flux (or direction of the impact; red denotes activation and blue inhibition) of the overall categories of pathways.

down-regulated DEG, several pathways related to hormone signaling were enriched, including insulin and growth hormone, as well as pathways associated with the immune system and other signaling pathways. Several of those KEGG pathways were the same as the one revealed to be impacted by DIA.

## Abundance of various miRs in exosomes isolated from bovine milk and porcine plasma

The complete dataset and statistical results of the miRs transcriptome from exosomes isolated from porcine plasma and bovine milk can be found in **S6 File**. The miRs sequencing in exosomes isolated from plasma generated 5,646,106±3,527,351 reads. Overall, there were 986 miRs (292 annotated to *Sus scrofa*) in the pig plasma exosomes and 316 (267 annotated to *Bos taurus*) miRs in the bovine milk exosomes that had at least one average read per million read counts (ARPM; **S6 File**).

The most abundant miRs in milk exosomes are well-annotated, such as miR-30a, miR30d, miR148a, and miR22 (**Fig 6A**). Among annotated *Sus scrofa* miRs, miR-181a, mir-191, and

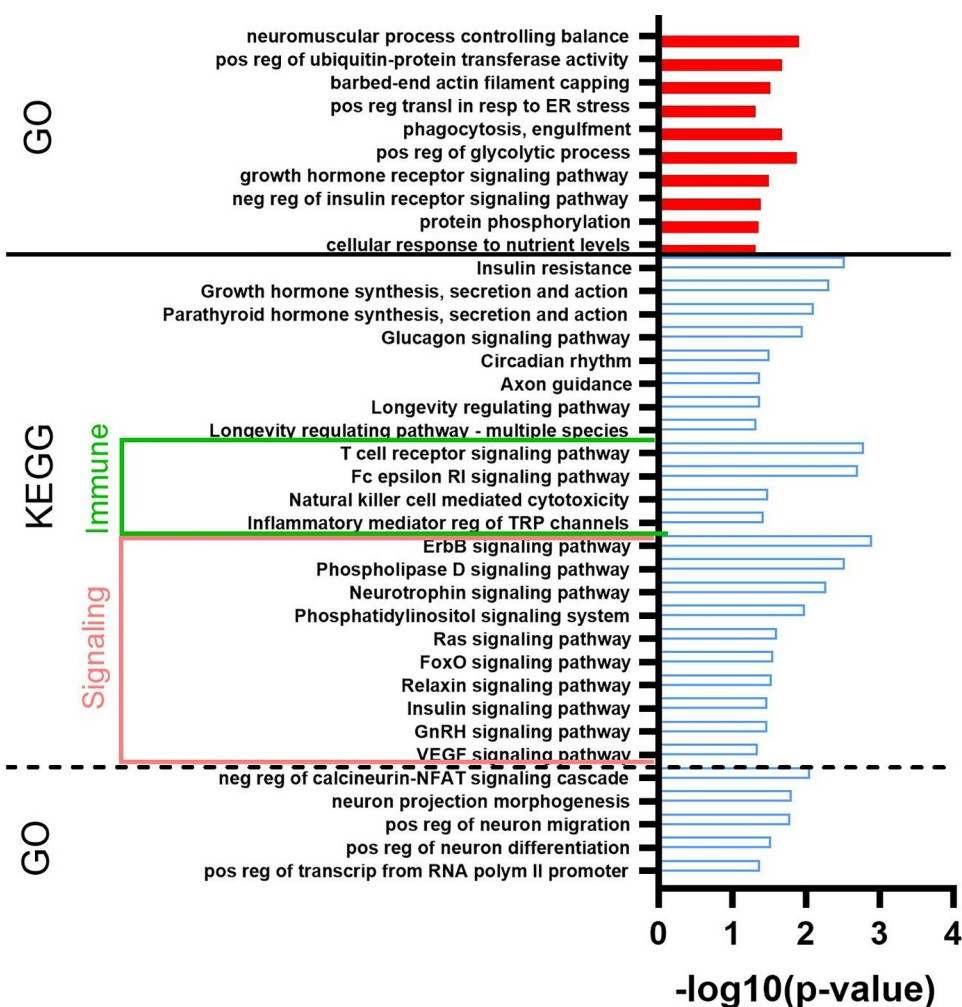

**Fig 5. Gene ontology process affected in adipose stem cells by consumption of milk.** Most enriched Gene Ontology process-related terms and KEGG pathways in up (red bars) and down (blue open bars) differentially expressed transcripts in adipose stem cells (ASC) from piglets supplemented with milk vs. ASC isolated from piglets supplemented with maltodextrin (control). The analysis was performed using DAVID. Horizontal bars denote the significance of enrichment.

miR92a were the most abundant in plasma exosomes (**Fig 6B**). However, we identified a large number of novel miRs, where the first eight most abundant miRs (between 62 and 96% of all miRs reads) were all novel with rr437876-1a-3p-miR (1 nt difference with miR183) being extremely abundant, accounting for 74% of all miR reads (**Fig 6C**).

## Comparison of miRs isolated from bovine milk vs. miRs isolated from porcine plasma

Our analysis identified 102 orthologous annotated miRs commonly detected in milk and plasma exosomes (**S6 File**). Most of those were proportionally more abundant (i.e., % of all miRs, P-value<0.05) in plasma obtained from porcine plasma exosomes compared to exosomes isolated from milk (**Fig 7**). Few miRs were proportionally more abundant in exosomes isolated from milk than those isolated from porcine plasma (**Fig 7**). Many orthologous miRs were uniquely present in each type of sample, with 138 miRs (57%) uniquely detected in milk.

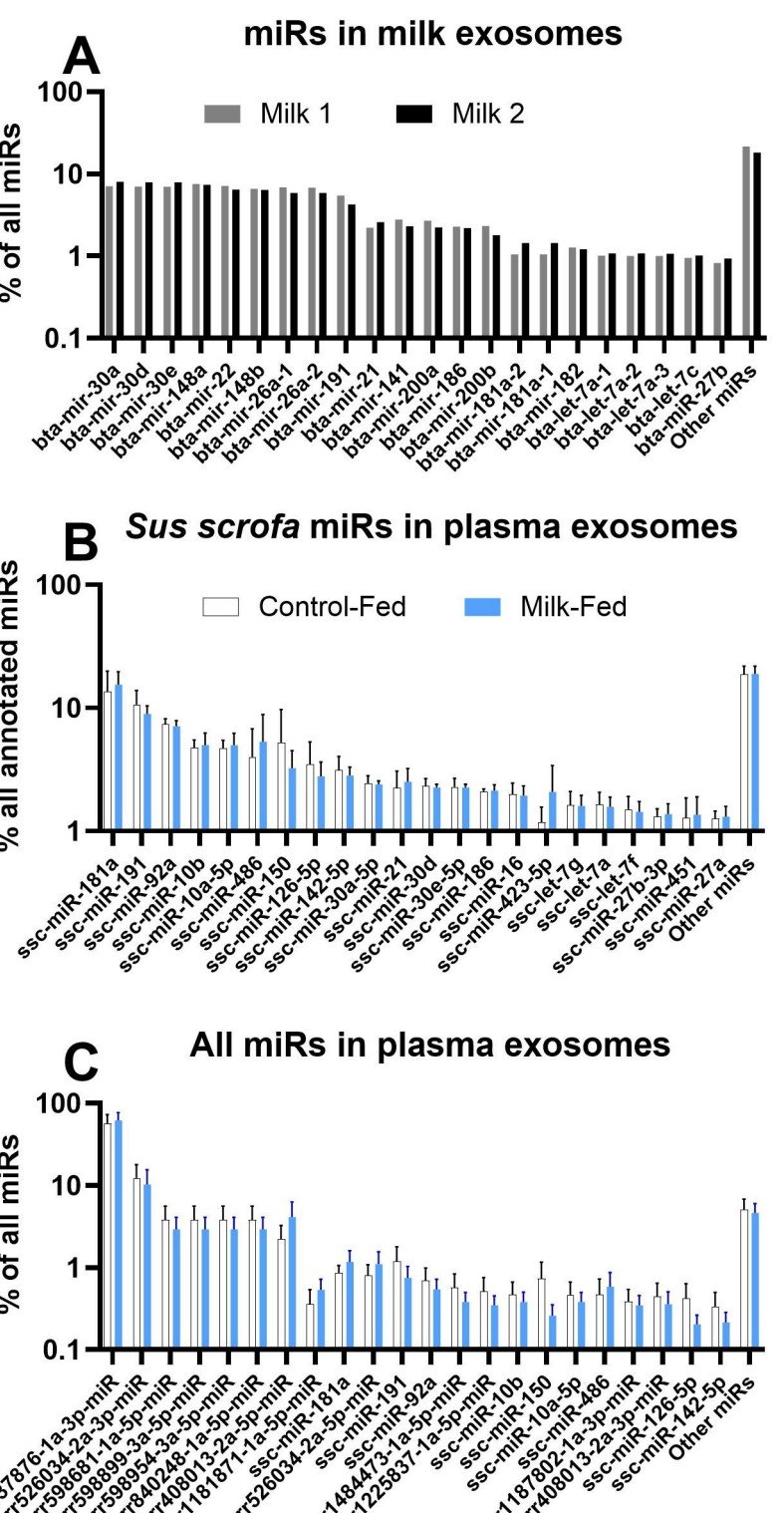

**Fig 6. miRs in bovine milk and porcine plasma.** Most abundant miRs as % of all measured miRs in bovine milk (A) and porcine plasma, both considering porcine-specific miRs (B) and all measured miRs (C). In B, the proportion of all measured miRs is reported. In C, the proportion of all *Sus scrofa*-specific miRs is reported.

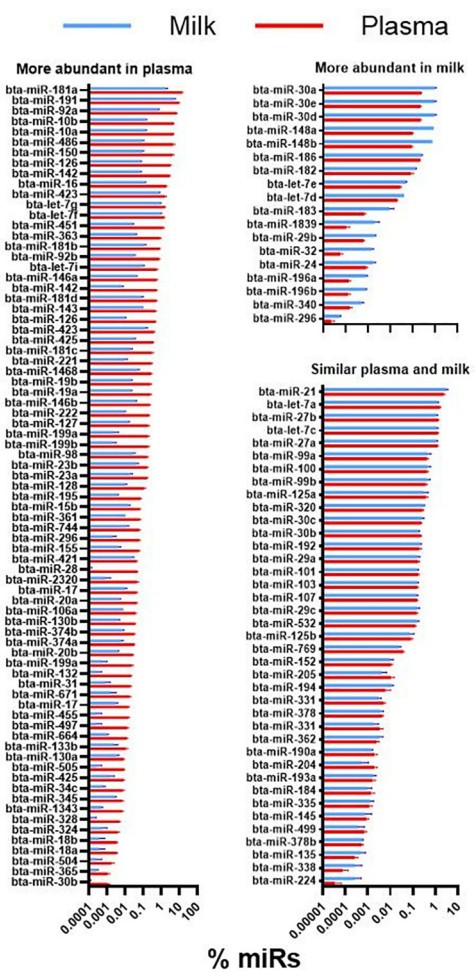

**Fig 7. miRs common between milk and porcine plasma.** miRs commonly present (as orthologous annotated miRs) in exosomes from bovine milk and porcine plasma. Reported are miRs that were proportionally more abundant (P<0.10) in plasma or milk or with a similar (P>0.1) percentage abundance between plasma and milk exosomes. The difference in % abundance of miRs was analyzed with a t-test between milk (n = 2) and plasma from the plasma of 6 pigs receiving milk.

Some of the most abundant miRs in exosomes isolated from milk, such as miR30a and miR30e, were also equally abundant in plasma (**Fig 7**). In contrast, other abundant miRs were uniquely present in milk exosomes, such as miR200a and miR22 (**Fig 8**).

## miRs in plasma exosome different between pigs supplemented with milk vs. maltodextrin

Among all miRs detected in porcine plasma exosomes, 38 miRs had a P-value<0.05. Still, only 3 had FDR <0.1 and were more abundant in exosomes isolated from the plasma of pigs fed milk vs. exosomes isolated from the plasma of pigs fed the control diet (**Table 1**).

The target genes of those miRs are included in **S7 File**. The combined target genes of all three miRs with a cumulative weighted context++ score of ≤-0.5 were analyzed for the enrichment of biological terms using DAVID (**S8 File**). The KEGG pathways and gene ontology biological process terms with the most significant enrichment are shown in **Table 2**. The Wnt pathway and the mitotic cell cycle were among the most enriched gene ontology terms.

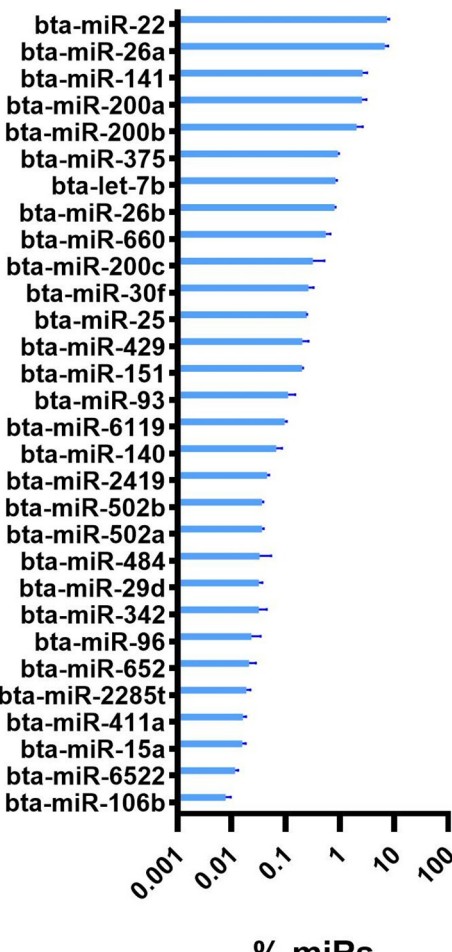

**Fig 8. Unique miRs in milk exosomes.** Thirty most abundant unique annotated miRs in milk exosomes (i.e., not detected in plasma exosomes).

None of the miRs found in bovine milk exosomes that were commonly (i.e., orthologous) present in bovine milk and porcine plasma were differentially expressed in the plasma exosomes isolated from the piglets supplemented with milk vs. the ones isolated from piglets supplemented with maltodextrin (i.e., control). (**S6 File**).

**Table 1. Porcine plasma exosome miRs with a statistically significant (FDR < 0.1) difference in abundance between pigs fed milk vs. pigs fed maltodextrin (control).**

| Originalannotation | Blast to bovine miRs | Milk vs. Control[1] | P-value | FDR | In milk exosomes[2] |
|---|---|---|---|---|---|
| rr804814-1a-5p-miR | bta-miR-449a | 13.0 | 2.43E-06 | 0.002 | N/D |
| ssc-miR-194b-5p | bta-miR-194b-5p* | 5.4 | 6.00E-05 | 0.024 | N/D |
| rr1265298-1a-5p-miR | not match | 29.1 | 0.00024 | 0.065 | N/D |

[1]fold difference in abundance

[2]N/D = not detectable

*1 nucleotide difference

**Table 2. DAVID enrichment analysis of genes targeted by the three miRs with significantly higher abundance in exosomes isolated from M compared to the exosomes isolated from C.** Shown is the most significant (EASE score<0.05) enriched gene ontology biological process terms and KEGG pathways in targets with a cumulative weighted context++ score of ≤-0.5 with TargetScan.

| Term | P-value |
|---|---|
| GO:0017157~regulation of exocytosis | 0.001 |
| GO:0034220~ion transmembrane transport | 0.004 |
| GO:0007268~chemical synaptic transmission | 0.007 |
| GO:0006810~transport | 0.011 |
| GO:0045930~negative regulation of mitotic cell cycle | 0.014 |
| GO:0010629~negative regulation of gene expression | 0.016 |
| GO:0014047~glutamate secretion | 0.016 |
| GO:0045608~negative regulation of auditory receptor cell differentiation | 0.021 |
| GO:0007386~compartment pattern specification | 0.028 |
| GO:0007409~axonogenesis | 0.032 |
| GO:0032527~protein exit from endoplasmic reticulum | 0.041 |
| GO:0016055~Wnt signaling pathway | 0.043 |
| GO:0014807~regulation of somitogenesis | 0.048 |
| GO:0042542~response to hydrogen peroxide | 0.050 |
| GO:0007269~neurotransmitter secretion | 0.050 |
| hsa04721:Synaptic vesicle cycle | 0.002 |
| hsa05033:Nicotine addiction | 0.040 |

## Potential target genes of miRs in milk among DEG in ASC

As we identified only three miRs in porcine plasma exosomes that were differentially abundant between pig-fed milk vs. maltodextrin, we performed an additional bioinformatic analysis to determine which miRs could affect the expression of DEG in ASC between the two groups of pigs and which among those could be found in high abundance in milk exosomes. The analysis revealed 459 unique miRs that could have affected the transcription of down-regulated DEG (S9 File). Among those 21 miRs were found in bovine milk including the miR30a and miR30e, the two most abundant miRs in milk exosomes, and miR21 and let7a, abundant in milk exosomes as well (Fig 6A), although all, except 4 (miR9, miR424, miR429, and miR484, all with a relatively low abundance in milk exosomes) were also present in porcine plasma exosomes.

**Table 3. miRs selection for testing exosome transfer from bovine milk to porcine plasma.**

| mRNA | Abundance in plasma | | | Abundance in milk |
|---|---|---|---|---|
| | *CTR* | *Milk* | *P-value* | |
| miR-29b | 21.9 | 10.2 | *0.21* | 26.4 |
| miR-200c | ND | ND | *N/A* | 287 |
| miR-1285 | 883 | 894 | *0.89* | ND |
| miR-148a | 1231 | 1030 | *0.92* | 11,412 |
| let-7b* | ND | ND | *N/A* | 1,134 |
| miR-200a | ND | ND | *N/A* | 3,008 |
| miR-22 | ND | ND | *N/A* | 9,348 |

*Bta-let-7b corresponds with Scc-let-7c (with one nucleotide difference); the abundance of this was 1648 for CTR and 1355 for Milk

We selected several exosomes to assess the transfer of exosomes from milk to porcine plasma (**Table 3**). The selection of miRs was based on a prior publication (miR29b and miR200c) [17], high abundance in milk exosomes but undetectability (let7b, miR200a, and miR22) or lower abundance (miR148a) in plasma exosomes, and/or present only in plasma exosomes (miR1285).

All miRs were detectable using RTqPCR, however, among milk-specific miRs, the detection was particularly good for let7b (contrary to the RNAseq data), while the detection was extremely low for the other miRs, even the ones detected in plasma exosomes using RNAseq (**Fig 9**). We detected milk bovine-specific transcripts in exosomes isolated from porcine plasma. However, the detection was extremely low (between 100 and >1000 fold lower than in exosomes isolated from bovine milk, **Fig 9**). The results reveal an increase in abundance of miR22 and miR29b in exosomes isolated from the pig plasma between 4 and 6 hours after milk consumption (**Fig 9**). No effect was observed for any of the other miRs or the milk-specific mRNAs *CSN3* and *LALBA*.

## Discussion

### Milk increases the number but not the adipogenic differentiation of ASC

To our knowledge, this is the first report on the role of milk consumption in ASC. Our data revealed that supplementing milk improved the quantity and proliferative capacity of ASC, although it did not affect their adipogenic differentiation capacity. Obesity is the consequence of an increased proliferation of ASC followed by their differentiation and lipid accumulation in response to a high-energy diet [3, 42]. In young humans, proliferation is the primary way to expand the adipose tissue [42]. An insufficient expansion of adipose tissue can induce ectopic fat accumulation that can trigger inflammation and insulin resistance [42]. The pigs in our experiment were not obese, and there was no difference in back and neck fat accumulation amongst treatment groups [23]. If the increase in proliferation we saw *in vitro* for pigs consuming milk is also true *in vivo*, then the increased proliferative capacity of ASC by feeding milk could be beneficial in preventing obesity. To determine if milk could improve the response to obesity, an experiment with a dietary high-energy model should be performed.

### Milk affects the transcripts related to inflammation, insulin signaling, and neuronal function in ASC

The effect of milk on the transcriptome of ASC is remarkable given that mesenchymal stem cells, besides affecting the formation of the adipose tissue, can have a broad impact on the body. Mesenchymal stem cells help to mitigate inflammation by interacting with the immune system [43] and participate in regenerating many tissues [44]; thus, a change in the transcriptome of those cells can have long-term and broad consequences. The bioinformatics analysis of the transcriptome revealed that the immune system and various signaling pathways, including insulin signaling, were the most critical functions affected by milk consumption compared with maltodextrin consumption.

In this study, milk supplementation compared with a maltodextrin solution had an anti-inflammatory effect by inhibiting the T cell receptor signaling that plays an essential role in a pro-inflammatory immune response, usually mitigated by mesenchymal stem cells [45–47]. Similarly, mesenchymal stem cells can down-regulate FcεRI signaling in mast cells, resulting in less inflammatory mediators' production [48]. The FcεRI signaling pathway in mast cells is associated with the activation of a pro-inflammatory response in the presence of antigens bound to IgE [48]. The FcεRI signaling was inhibited in the ASC of pigs supplemented with

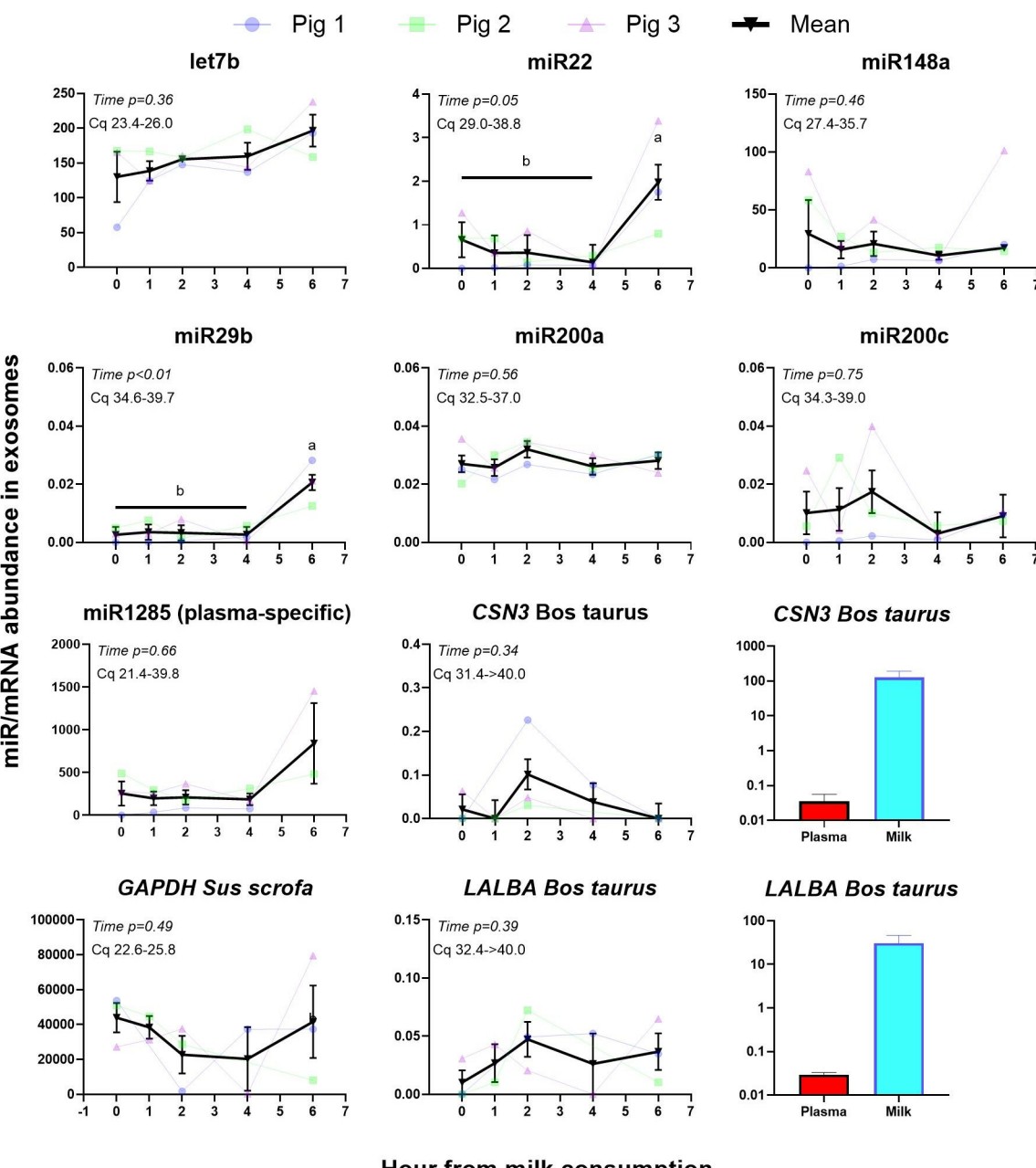

**Fig 9. Assessment of vertical transfer of milk exosomes in circulation of piglets.** Transcription abundance of selected miRs expected to be abundantly or uniquely present in bovine milk exosomes (let7b, miR22, miR48a, miR29b, miR200a and miR200c), miR1285 expected to be present only in porcine plasma exosomes, bovine milk-specific mRNA (*CSN3* and *LALBA*), and a porcine housekeeping gene (*GAPDH*) present in exosomes isolated from porcine plasma before and during the 6 hours after milk consumption. The P-value of the time effect is reported, and different letters denote significant effect (P<0.05) between the various time points. The range of the Cq (crossing point, a.k.a. cycle to threshold).

milk compared with a maltodextrin solution. Inflammation is important in the context of obesity because it has been strongly linked to insulin resistance [49, 50]. With the increase in pro-inflammatory cytokine secretion by adipose cells, the expression of insulin receptor substrate and glucose transporter 4 is decreased, inducing insulin resistance [47]. Our data support this link as it reveals an inhibition of insulin resistance in ASC from pigs supplemented milk

compared with a maltodextrin solution. Interestingly, milk supplementation also appears to inhibit insulin and growth hormone signaling. This would contrast with the apparent decrease in insulin resistance. One of the central genes down-regulated by milk supplementation on ASC related to insulin signaling is the gene coding for phosphoinositide 3-kinase (PI3K) proteins. The PI3K is a crucial protein in both adipogenesis [51] and osteogenesis [52] of mesenchymal stem cells and plays a central role in insulin signaling where the disruption of the PI3k/AKT signaling is associated with insulin resistance and obesity [53]. Thus, the downregulation of PI3K would suggest that milk consumption would decrease the adipogenesis capacity of ASC, although we did not observe such a phenotype.

One pathway that was not highly impacted by treatment but is still of interest is the tumor necrosis factor (TNF) signaling pathway. The TNF signaling pathway is of interest due to its negative association with adipogenesis and positive association with an obese phenotype [54]. TNFα has been shown to repress Peroxisome Proliferator-Activated Receptor γ (PPARγ), one of the leading transcription factors responsible for adipogenic differentiation [54]. In addition, TNFα and other pro-inflammatory cytokines play a role in dysregulating the cyclic guanosine monophosphate signaling pathway (cGMP) necessary for differentiating ASC [54]. The down-regulation of the TNF signaling pathway, along with the upregulation of the cGMP pathway in pigs consuming milk, would suggest increased adipogenesis of the ASC. However, our data only showed a numerically larger quantity of ASC in pigs consuming milk vs. pigs consuming maltodextrin in passage 1 ASC. TNFα is also involved in inducing insulin resistance [55]; thus, a potential decrease of TNFα signaling by the decrease of inflammatory pathways seen in pigs supplemented with milk might benefit insulin sensitivity. A reduction in insulin resistance in ASC in pigs supplemented with milk is also supported by inhibiting the Mitogen-Activated Protein Kinase 1 (MAPK) signaling pathway [56]. Overall, the decrease in inflammatory-related pathways and MAPK supports a potential reduction in insulin resistance and risk of type II diabetes in ASC of pigs consuming milk compared with a maltodextrin solution.

Among neuronal system pathways, neurotrophin signaling was inhibited by milk consumption. This is important because ASC can differentiate into various cell lines, including neurons [57]. This pathway promotes neuronal differentiation along with neuron survival, meaning the down-regulation of this pathway could result in less neuronal differentiation [58]. This is further supported by the putative target genes of the miRs in plasma exosomes that were more abundant in pigs fed milk vs. a control diet. According to the bioinformatics analysis, this inhibition of the neurotrophin signaling pathway in pigs supplemented with milk could negatively affect pathways such as Wnt signaling and cell cycle. The Wnt pathway is important for cell proliferation and differentiation and has been shown to inhibit adipogenesis when overexpressed [56]. The Wnt pathway has also been shown to increase neuronal differentiation when stimulated by neurotrophins [59]. Overall, this data would suggest that the ASC from the control piglets would be more prone to neurogenesis and less prone to adipogenesis when compared to the ASC from the piglets consuming milk. This could have important implications for using mesenchymal stem cells for neuronal disorders/regeneration [60]. In addition, by ASC from pigs consuming milk showing down-regulation of genes associated with the negative regulation of the mitotic cell cycle, we would expect a higher proliferation [61], as we observed.

The ErB signaling and VEGF signaling pathways, both associated with ASC proliferation and differentiation, were inhibited in the ASC of pigs consuming milk [62, 63]. In addition, the Hedgehog pathway, shown to decrease adipogenesis in human ASC, was up-regulated in the same group [64].

It has been previously argued that milk consumption, especially its high protein content, could hyperactivate mTORC1 and cause obesity [7, 21, 65]. In our analysis, the mTOR pathway was not highly impacted and was inhibited overall in ASC isolated from pigs consuming

milk (see S4 and S5 Files). Our data contrasts with a previous study conducted in mice by Bar Yamin et al. [66]. In that study, mice fed whole milk had a decrease in AMPK (but an increase in pAMPK), one of the major energy-sensing systems that inhibit mTOR signaling, and a concomitant increase in phosphorylation and nuclear presence of S6K1, a downstream of mTORC1. The nuclear S6K1 can block the expression of WNT genes and, as a consequence, increase adipogenesis and proliferation of ASC by increasing the expression of the two master regulators PPARγ and CEBP/alpha [67–70]. In our study, we observed an increased proliferation of ASC from pigs fed with milk; however, we did not observe higher adipogenesis of ASC isolated from pigs fed milk, and the transcription of the various WNT isoforms, *PPARG*, and *CEBPA*, was not affected by feeding milk. Furthermore, the pigs fed milk did not accumulate more fat than the control animals [23]. Despite our data not supporting a higher activation of mTOR in ASC response to feeding milk, we cannot exclude that mTOR was affected by feeding milk to pigs. This could be determined by measuring mTOR activation on ASC, which was, however, beyond the scope of our study.

The study by Bar Yamin et al. also noted a decrease in insulin signaling for mice consuming whole milk [66]. This was similar to the results reported by Hoppe et al. [71], where they found an increase in insulin secretion and resistance in children consuming 53g of protein from whole milk compared to a control group. Our bioinformatic analysis suggested down-regulation of insulin resistance-related pathways in ASC of pigs consuming milk compared with those consuming a maltodextrin solution. The differences compared to the prior study are two-fold: 1) our study design had control pigs supplemented with isocaloric maltodextrin, while the studies mentioned above had diets with and without milk; 2) the above studies look into the adipose tissue while we isolated and studied ASC.

For the miRNA, miR-449a and miR-4251, both found in greater abundance in pigs supplemented with milk, have mainly been studied in the context of cancer treatment, with miR-449a regulating proliferation [72]. This contrasts with the increase in proliferation observed in ASC of pigs consuming milk. In addition, miR-194b has been linked to decreased adipogenesis through the deactivation of chicken ovalbumin upstream promoter-transcription factor II (COUP-TFII), essential for mesenchymal stem cell differentiation to adipocytes [73]. Thus, our study does not support those prior observations, as milk consumption did not affect ASC adipogenesis. One possibility for this difference is that plasma exosomal miRs are not acting on the ASC. It has been previously reported that miRNA secreted in microvesicles from ASC can influence differentiation [74]. For our study, we may have observed more differentially expressed miRNA had we isolated miRNA from the subcutaneous adipose tissue in addition to the plasma.

Overall, the differentially expressed pathways in the ASC of pigs supplemented with milk tended to be associated with a down-regulation in inflammation and insulin resistance and an up-regulation in adipogenesis and proliferation. This was at odds with the pathways associated with the more abundant miR isolated from milk-fed porcine plasma and the lack of difference in adipogenesis between the two treatment groups.

## miRs from milk are not transferred to circulation

Previous studies have found various miRs to be highly abundant in bovine milk exosomes. These miRs include miR-200c, miR-26a, and miR-148a amongst others [75–77]. In this study, the top ten most abundant miRs in milk, including miR30, miR148, miR22, miR191, and miR41, have not previously been reported as highly abundant in bovine milk exosomes [75–77]. Of these miRs, miR19, miR30, and miR148 were among the most abundant annotated miRs in pig plasma exosomes. This finding can hinder a proper evaluation of the transfer of

miRs from the intestine to circulation (i.e., vertical transfer) due to the high conservation of miRs sequences between species, as previously argued [78].

Several researchers provided data supporting a vertical transfer of bovine milk exosomes (and/or their miRs) affecting peripheral tissues, including the brain [17, 18, 79–84]. Recently, vertical transmission of miRs from bovine colostrum in pig plasma using xenomiRs was reported [85]. The vertical transfer of miRs is not without controversy. A reanalysis using microarray and RNA sequencing of miRs from plasma samples collected in humans after milk consumption, where an evident increase of milk-specific miRs was observed [17], failed to confirm the data [86]. It is worth noting that, for that study, samples arrived without dry ice but were evaluated and found not to have signs of degradation. A recent literature review revealed several studies where there was no evidence of uptake of exosomes from milk (or other food) into circulation [87]. Our pig plasma miRs profiles did not support any transfer of bovine milk miRs into the circulation of piglets when blood was collected according to previous study protocols. Based on those findings, we hypothesized that we could have missed the window of uptake of miRs, as the pig's blood was collected >6 hours after feeding the milk. Oligonucleotides have a relatively short half-life in circulation, and the absorption of several compounds by the intestine is relatively rapid, with a peak during the first 2 hours after a meal [88]. In a human study fed uridine, the peak presence of uridine in blood was during the first 4-hour post-meal [89]. Therefore, we proceeded with the second study to assess if miRs from milk can be vertically transferred into porcine circulation, following methodologies that successfully captured milk miRs vertical transfer in humans [17]. The high abundance of milk-specific mRNA in milk exosomes, such as *LALBA* and *CSN3* [90], and a relatively low identity in the sequence of those mRNA between pigs and bovines (81.5 and 78% identical alignment, respectively) allow for bovine-specific primers to be designed while avoiding confounding effects when measuring the highly conserved miRs [78]. Because of the above, we measured those mRNA to assess the presence of a vertical transfer of exosomes from milk to circulation, as those should be carried with the exosomes as the miRs. We detected those two mRNA but at an extremely low level and with no noticeable increase after milk consumption. Thus, despite our efforts, we could not detect any vertical transfer of miRs or mRNA from bovine milk to porcine circulation. While these findings do not support exosome vertical transfer amongst species, they reveal that miRs are conserved when comparing profiles of pig plasma to cow milk.

So, how can we best explain our findings? Our experiment may have missed the window to capture any milk-derived miRs in pig plasma. Lin et al. [91] detected a peak in milk-specific miRs in porcine plasma between 3 and 6 days of feeding milk but with a level similar to before feeding milk at 12 days. This is relevant, as in our first experiment, we collected blood from the piglets only once at the end of one month of feeding milk. Still, in our second experiment, we collected blood to measure miRs after six days of feeding milk, corresponding to the maximal period of milk-specific miRs detected by Lin et al. [91]. Despite this, we did not detect any miRs.

The portion of our study that explored the vertical transfer of miRs was based on Baier et al. [17] which used adult humans. However, the available studies carried out in piglets by Lin et al. [91] and Weil et al. [85] used neonatal piglets, while we used between 5 and 9-week-old piglets. More recent studies on the vertical transfer of exosomes [82, 83] were performed in nursing mice pups. It has been well-established that the intestine of neonatal mammals is more porous, allowing for the absorption of relatively large proteins until the maturation of the intestinal epithelium, around 6–9 months of age in humans [92–94]. In piglets, the intestinal closure is very fast, with most of the closure happening 24 hours post-birth with a complete closure before 6 weeks of age [95], although the uptake of large molecules by the intestinal

epithelia is still present after 24h, no significant transfer of those molecules into circulation is observed [96, 97]. The colostrum consumed in the early post-birth stage helps close the intestine to the transfer of large molecules, with a possible preeminent role of extracellular vesicles, including exosomes [98, 99]. Thus, it is possible that the positive vertical transfer of milk miRs observed in the above-cited studies was due to the porous intestine of nursing animals. In contrast, studies in humans, in which a vertical transfer of miRs was observed, were carried out in adults [17, 100]. The milk consumed was relatively large (from 0.25 to 1 L). Large amounts of lactose might not be promptly digested, upsetting the intestine and inducing a leaky gut, where the impaired epithelial barrier allows large particles, such as exosomes, to be transferred [101]. In piglets, weaning stress can compromise intestinal health, affecting the absorption capacity of the intestine and, in worst cases, destroying the intestinal barrier and compromising the growth of the animals [102]. The piglets used in our study were all healthy and had no apparent gastrointestinal issues [23]. Thus, we assumed that the stress of weaning those animals did not compromise their health or, as a consequence, the results of our study. Except for a few instances of loose feces after the introduction of lactose or maltodextrin supplementation [23]. Those were not serious and resolved fast, indicating that the piglets could digest lactose and that no intestinal issues were present in the animals during the study.

Unlike all the above-cited studies where exosome transfer was assessed, we isolated exosomes from the plasma before measuring miRs. In contrast, the other studies measured miRs directly in plasma or serum without isolating exosomes. As there are different types of extracellular cargos [103], it is possible that our isolation excluded vesicles that carried milk-derived miRs. Thus, it is possible that milk-specific miRs detected in circulation in the above studies are not from exosomes but from other vesicles.

## Limitations of the study

As determined in our prior paper [23], the accumulation of fat and the distribution of adipocyte size can respond differently to feeding milk between pig breeds. Thus, our findings, especially concerning ASC, might be limited to using the Duroc-Berkshire breed. Using ultracentrifugation only in milk and not in plasma could have confounded the findings. However, the distribution size of exosomes was similar between milk and plasma.

We isolated ASC from the subcutaneous adipose tissue; however, visceral fat can be more relevant for male humans, as fat accumulation is more significant in visceral fat depot than in the subcutaneous adipose tissue [104]. The two fat depots respond differently to insulin, as observed in the Munich MIDY pig model [105]. The ASC from the two depots are very similar in humans, as demonstrated via a whole transcriptomic analysis by Baglioni et al. [106]. Despite this, as the authors of that study also suggested, we recognized that the use of visceral ASC might have been more relevant to be investigated in our study.

Using isocaloric maltodextrin supplementation instead of an isocaloric and isonitrogenous supplementation can be considered a significant limitation considering the role of amino acids, especially branched-chain amino acids, on adipogenesis via mTOR [7]. Even though our data do not suggest an effect of milk consumption on mTOR of ASC, the use of an isonitrogenous, or even better, an amino acids-match control diet, would have allowed us to investigate more clearly the effect of milk. The reason for using only an isocaloric supplement was based on the aim to mimic the "real world" situation where children replace milk consumption with sugar-sweetened beverages, as previously discussed [107].

## Conclusions

Our data only partly supported the original hypothesis of the effect of milk on ASC via changing the miRs in exosomes. The milk did affect the ASC, phenotypically and at the transcriptomic level; however, the changes observed were not due to miRs from milk exosomes. This was further supported by the time course study, where we failed to demonstrate a vertical transfer of miRs from milk exosomes to circulation, as initially hypothesized. The transcriptomic data on ASC revealed an effect of milk on decreasing inflammatory-related pathways, insulin signaling, and neuronal-related pathways when compared with consuming a maltodextrin solution. The significance of our findings related to childhood obesity is not completely clear, especially regarding the effect on insulin signaling. Finally, milk contains many bioactive compounds that could have affected the transcriptome of the ASC, including branched-chain amino acids [108–110]; however, the investigation of those was beyond the aim of the present research.

## Supporting information

**S1 File. CellProfiler pipeline used to analyze the number of adipocytes and the content of lipid droplets in the present manuscript.**
(CPPROJ)

**S2 File. Trough, adipose stem cells (ASC) transfection and differentiation, exosomal markers, and Volcano plot of differentially expressed genes in ASC.** *Fig 1*. Trough used to provide milk to the piglets; *Table 1* and *Fig 2*. Data and results of the transfection of adipose stem cells with various transfection agents; *Figs 3* and *4*. Images of the adipose stem cells differentiated in adipocytes and osteocytes; *Fig 5*. Analysis of exosomal markers from exosomes isolated form milk and plasma of pigs; *Fig 6*. Transcript abundance of mesenchymal and porcine-specific adipose stem cells markers in ASC; *Fig 7*. Volcano plot of the DEG in between ASC from pigs supplemented with milk vs. ASC of milk supplemented with isocaloric maltodextrin solution.
(DOCX)

**S3 File. Complete results of the transcriptome analysis of adipose stem cells.**
(XLSX)

**S4 File. Details results of KEGG pathway analysis by the Dynamic Impact Approach of the adipose stem cells transcriptome of piglets receive milk or maltodextrin as supplement.**
(XLSX)

**S5 File. Details results of the enrichment analysis by DAVID of differentially expressed genes between adipose stem cells of piglets receiving milk vs. adipose stem cells of piglets receiving maltodextrin as a supplement.**
(XLSX)

**S6 File. Overall results of miRNA analysis of bovine milk and porcine plasma and statistical results of the comparison of the miRNA between exosomes isolated from piglets fed milk vs. piglets fed maltodextrin.**
(XLSX)

**S7 File. Targets of miRNA that were differently abundant in exosomes isolated from piglets fed milk vs. piglets fed maltodextrin.**
(XLSX)

**S8 File. Enrichment analysis of functions by DAVID of target genes of the miRNA that were differently abundant in exosomes isolated from piglets fed milk vs. piglets fed**

**maltodextrin.**
(XLSX)

**S9 File. miRNA targeting genes that were differentially expressed in adipose stem cells isolated from piglets fed milk vs. adipose stem cells isolated from piglets fed maltodextrin.**
(XLSX)

## Acknowledgments

The authors thank the following students for their help during the experiment: Heaven Roberts, Sarah Akers, Cassie Penix, Emily Sahagun, Randi Wilson, Tamay Guevara, Minda Newhouse, Katie White, Eric Tam, Catalina Tello, Enrique Perez, Kaitlyn Vander Pas, Nina Enos, Emilie Peterson, Nicolas Rivero, Winnie Luo, Nicolas Aguilera, and Sebastiano Busato for their help with feeding the piglets and sample collection.

## Author Contributions

**Conceptualization:** David Hendrix, Massimo Bionaz.

**Data curation:** David Hendrix, Massimo Bionaz.

**Formal analysis:** Katherine Swanson, Jimmy Bell, David Hendrix, Duo Jiang, Brandon Batty, Melanie Hanlon, Massimo Bionaz.

**Funding acquisition:** David Hendrix, Michelle Kutzler, Massimo Bionaz.

**Investigation:** Katherine Swanson, Michelle Kutzler, Brandon Batty, Melanie Hanlon, Massimo Bionaz.

**Methodology:** Katherine Swanson, Jimmy Bell, David Hendrix, Massimo Bionaz.

**Project administration:** Massimo Bionaz.

**Resources:** David Hendrix, Massimo Bionaz.

**Software:** David Hendrix.

**Supervision:** Massimo Bionaz.

**Validation:** Massimo Bionaz.

**Visualization:** Massimo Bionaz.

**Writing – original draft:** Katherine Swanson, Massimo Bionaz.

**Writing – review & editing:** Jimmy Bell, David Hendrix, Duo Jiang, Michelle Kutzler, Brandon Batty, Melanie Hanlon, Massimo Bionaz.

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
