## [Decision Letter · Decision Letter 0]

7 May 2024

PONE-D-24-14376Bovine Milk Consumption Affects the Transcriptome of Porcine Adipose Stem Cells: do Exosomes Play any Role?PLOS ONE

Dear Dr. BIONAZ,

Thank you for submitting your manuscript to PLOS ONE. After careful consideration, we feel that it has merit but does not fully meet PLOS ONE’s publication criteria as it currently stands. Therefore, we invite you to submit a revised version of the manuscript that addresses the points raised during the review process.Please find below the reviews of two experts in the field, which both recommended improvements to make the manuscript publication-ready.

We look forward to receiving your revised manuscript.

Kind regards,

Wilfried A. Kues, Ph.D.

Academic Editor

PLOS ONE

Journal Requirements:

"This work was supported by the USDA National Institute of Food and Agriculture (Washington, DC) Agriculture and Food Research Initiative Exploratory, grant # 2015-67030-23872 to MB, DH, and MK, along with the National Needs Graduate Fellowships, Grant #2014-38420-21800 (MB was among the co-PI and supported the PhD of KS). "

 "USDA National Institute of Food and Agriculture (Washington, DC) Agriculture and Food Research Initiative Exploratory, grant # 2015-67030-23872, along with the National Needs Graduate Fellowships, Grant #2014-38420-21800"

"USDA National Institute of Food and Agriculture (Washington, DC) Agriculture and Food Research Initiative Exploratory, grant # 2015-67030-23872, along with the National Needs Graduate Fellowships, Grant #2014-38420-21800"          

5. Please amend your authorship list in your manuscript file to include author "Michelle Kutzler".

Reviewers' comments:

Reviewer's Responses to Questions

**Comments to the Author**

1. Is the manuscript technically sound, and do the data support the conclusions?

Reviewer #1: No

Reviewer #2: No

2. Has the statistical analysis been performed appropriately and rigorously? 

Reviewer #1: N/A

Reviewer #2: Yes

3. Have the authors made all data underlying the findings in their manuscript fully available?

Reviewer #1: No

Reviewer #2: Yes

4. Is the manuscript presented in an intelligible fashion and written in standard English?

Reviewer #1: No

Reviewer #2: No

5. Review Comments to the Author

Reviewer #1: 1. The abstract is not fully described, the background is not sufficient to support the current study, and the important research objectives and significance are missing. Please rewrite.

2. Introduction (lines 43-51): The focus of your study is not highlighted, so it is suggested to describe the function of ASC in detail.

3. Introduction (lines 52-62): The logic is unclear, so it is suggested to organize the logical structure.

4. Lines 65-69: The reference citations are not standardized, [11,12] appear twice in consecutive sentences.

5. Lines 80-82: You have not detected the expression of miRs in ASC and have not analyzed the transfer of miRs in exosomes in ASC, so how did you arrive at this conclusion? It is suggested to supplement experimental data on the transfer of miRs in exosomes in ASC.

6. Line 180: The reference format is incorrect.

7. Line 361: Fig. 1A and B have no significant differences, so it is suggested to describe the three indicators separately in detail.

8. Line 369: Exosomes? should be described in detail for the title.

9. Line 388-394: The description of Fig. 3 is too brief and lacks important results.

10. Line 410-420: The up-regulated and down-regulated genes are not found in Fig. 5, did the author miss labeling them?

11. Fig. 7 and Fig. 9 lack a number for each result.

12. Line 495-498: No results for body weight are found, how did the conclusions come about?

13. Line 515: What is FcԐRI signaling?

14. Line 594-596: Reference [77] has already explained that there is no evidence to suggest that exosomes from milk are taken up and enter the bloodstream. Why do you detect the transfer of exosomes in plasma and suggest referencing relevant literature and supplementary experiments to explain this?

15. Line 616-646: Please carefully check the citation format of references.

16. The entire discussion section lacks clarity in its logical structure. It is strongly recommended that the author rewrites the discussion section, paying attention to the logical structure, properly cites references, and fully discusses the results of their own research.

Reviewer #2: 1- As noted in both MISEV2018 and MISEV2023 guidelines as well as many other scientific reports, using the exosome term should be limited to experiments that have shown biogenesis of exosomes routed from MVB. Therefore, based on the methodology of this manuscript it is the wrong term. Since the others have only shown size distribution, they cannot even use extracellular vesicle term either. The identity of what has been administered to cells should be clearly investigated and proper terms should be used. Based on MISEV guidelines, marker expression and morphology should be checked.

2- The manuscript is full of language mistakes which made reading difficult and maybe impossible. For example, “Exosomes are around 100 nm in diameter microvesicles that work... “

3- The rationality of the work is not strong. Old references were used and many sentences don’t have proper references.

4- The titles in the method part were very weird. For example, exosome isolation is under “Animals and experimental design”

6. PLOS authors have the option to publish the peer review history of their article (what does this mean?). If published, this will include your full peer review and any attached files.

Reviewer #1: No

Reviewer #2: No

---

## [Author Response · Author response to Decision Letter 0]

22 Jul 2024

AU: we checked, and the format should be correct.

"This work was supported by the USDA National Institute of Food and Agriculture (Washington, DC) Agriculture and Food Research Initiative Exploratory, grant # 2015-67030-23872 to MB, DH, and MK, along with the National Needs Graduate Fellowships, Grant #2014-38420-21800 (MB was among the co-PI and supported the PhD of KS). "

 "USDA National Institute of Food and Agriculture (Washington, DC) Agriculture and Food Research Initiative Exploratory, grant # 2015-67030-23872, along with the National Needs Graduate Fellowships, Grant #2014-38420-21800"

Authors: We remove the references to the grant agencies as indicated.

"USDA National Institute of Food and Agriculture (Washington, DC) Agriculture and Food Research Initiative Exploratory, grant # 2015-67030-23872, along with the National Needs Graduate Fellowships, Grant #2014-38420-21800" 

AU: added in the cover letter as indicated.

AU: we have submitted all the data not previously put in repositories (i.e., RNAseq) into the DANS Data Station Life Sciences. The dataset is under review and should be available soon.

5. Please amend your authorship list in your manuscript file to include author "Michelle Kutzler".

AU: thanks for catching that! Now Michelle has been added.

Reviewers' comments:

Reviewer's Responses to Questions

Comments to the Author

1. Is the manuscript technically sound, and do the data support the conclusions?

Reviewer #1: No

Reviewer #2: No

Not really sure how to reply to this. Seems subjective. 

2. Has the statistical analysis been performed appropriately and rigorously?

Reviewer #1: N/A

Reviewer #2: Yes

3. Have the authors made all data underlying the findings in their manuscript fully available?

Reviewer #1: No

Reviewer #2: Yes

4. Is the manuscript presented in an intelligible fashion and written in standard English?

Reviewer #1: No

Reviewer #2: No

Please refer to edited manuscript. 

5. Review Comments to the Author

Reviewer #1: 1. The abstract is not fully described, the background is not sufficient to support the current study, and the important research objectives and significance are missing. Please rewrite.

AU: The abstract has been edited to clarify the objectives and conclusions

2. Introduction (lines 43-51): The focus of your study is not highlighted, so it is suggested to describe the function of ASC in detail.

AU: The focus of the study is outlined at the end of the introduction. The function of ASC in obesity is described in lines 53-55. 

3. Introduction (lines 52-62): The logic is unclear, so it is suggested to organize the logical structure.

AU: The paragraph has been reorganized for clarity

4. Lines 65-69: The reference citations are not standardized, [11,12] appear twice in consecutive sentences.

AU: This is due to both studies being referenced in both sentences

5. Lines 80-82: You have not detected the expression of miRs in ASC and have not analyzed the transfer of miRs in exosomes in ASC, so how did you arrive at this conclusion? It is suggested to supplement experimental data on the transfer of miRs in exosomes in ASC.

AU: We have modified the sentence to address this point. However, we did not analyze the miRs in ASC. This would have been a good idea….In any case, based on the “vertical transfer” hypothesis we should detect an effect on the cell transcriptome. 

6. Line 180: The reference format is incorrect.

AU: Thanks for catching it! Now it is addressed.

7. Line 361: Fig. 1A and B have no significant differences, so it is suggested to describe the three indicators separately in detail.

AU: We are not sure about this comment. The Figures 1A had a tendency while the Figure 1B was significant.

8. Line 369: Exosomes? should be described in detail for the title.

AU: Edited in text

9. Line 388-394: The description of Fig. 3 is too brief and lacks important results.

AU: Edited in text to further elaborate on Fig. 3. 

10. Line 410-420: The up-regulated and down-regulated genes are not found in Fig. 5, did the author miss labeling them?

AU: The legend should provide the necessary explanation, as those are not up-and down-regulated genes but are the GO terms from the analysis of up- and down-regulated DEG. We checked the legend for Figure 5 and appeared to be clear.

11. Fig. 7 and Fig. 9 lack a number for each result.

AU: Each figure has a title that indicates the parameter shown. The legend should provide further details. We can add letters to indicate each figure, if needed, but it appears to increase the complexity of the figure rather than helping. 

12. Line 495-498: No results for body weight are found, how did the conclusions come about?

AU: Body weight was reported in a previous paper referenced in the beginning of the methods section. 

13. Line 515: What is FcԐRI signaling?

AU: This pathway is now further described in the text. 

14. Line 594-596: Reference [77] has already explained that there is no evidence to suggest that exosomes from milk are taken up and enter the bloodstream. Why do you detect the transfer of exosomes in plasma and suggest referencing relevant literature and supplementary experiments to explain this?

AU: Our study did not detect the transfer of exosomes. We sought to further explain changes in the ASC transcriptome by determining whether vertical transfer of exosomes is possible, but our data do not support that theory. 

15. Line 616-646: Please carefully check the citation format of references.

AU: Any errors in citation format have been fixed.

16. The entire discussion section lacks clarity in its logical structure. It is strongly recommended that the author rewrites the discussion section, paying attention to the logical structure, properly cites references, and fully discusses the results of their own research.

AU: We have edited the text to improve the logical flow of the discussion and remove unnecessary information. We have also fixed all citations. 

Reviewer #2: 1- As noted in both MISEV2018 and MISEV2023 guidelines as well as many other scientific reports, using the exosome term should be limited to experiments that have shown biogenesis of exosomes routed from MVB. Therefore, based on the methodology of this manuscript it is the wrong term. Since the others have only shown size distribution, they cannot even use extracellular vesicle term either. The identity of what has been administered to cells should be clearly investigated and proper terms should be used. Based on MISEV guidelines, marker expression and morphology should be checked.

AU: Additional work has been added to show confirmation of exosomes markers. This has been added to the methods and results. 

2- The manuscript is full of language mistakes which made reading difficult and maybe impossible. For example, “Exosomes are around 100 nm in diameter microvesicles that work... “

AU: The manuscript has been edited to make reading easier. 

3- The rationality of the work is not strong. Old references were used and many sentences don’t have proper references.

AU: The manuscript has been edited. We have checked the references, and they all appear proper. 

4- The titles in the method part were very weird. For example, exosome isolation is under “Animals and experimental design”

AU: Titles have been clarified in the manuscript.

---

## [Decision Letter · Decision Letter 1]

23 Aug 2024

PONE-D-24-14376R1Bovine Milk Consumption Affects the Transcriptome of Porcine Adipose Stem Cells: do Exosomes Play any Role?PLOS ONE

Dear Dr. BIONAZ,

Thank you for submitting your manuscript to PLOS ONE. After careful consideration, we feel that it has merit but does not fully meet PLOS ONE’s publication criteria as it currently stands. Therefore, we invite you to submit a revised version of the manuscript that addresses the points raised during the review process. Please have a close look on the points brought up by to experts in the field (below).

We look forward to receiving your revised manuscript.

Kind regards,

Wilfried A. Kues, Ph.D.

Academic Editor

PLOS ONE

Reviewers' comments:

Reviewer's Responses to Questions

**Comments to the Author**

1. If the authors have adequately addressed your comments raised in a previous round of review and you feel that this manuscript is now acceptable for publication, you may indicate that here to bypass the “Comments to the Author” section, enter your conflict of interest statement in the “Confidential to Editor” section, and submit your "Accept" recommendation.

Reviewer #3: All comments have been addressed

Reviewer #4: (No Response)

2. Is the manuscript technically sound, and do the data support the conclusions?

Reviewer #3: Yes

Reviewer #4: Partly

3. Has the statistical analysis been performed appropriately and rigorously? 

Reviewer #3: Yes

Reviewer #4: Yes

4. Have the authors made all data underlying the findings in their manuscript fully available?

Reviewer #3: Yes

Reviewer #4: Yes

5. Is the manuscript presented in an intelligible fashion and written in standard English?

Reviewer #3: Yes

Reviewer #4: Yes

6. Review Comments to the Author

Reviewer #3: This paper present enough data, which may be interesting to readers. I think this ms had been well revised according to comments of reviewers. I deem it is acceptable, though it is of the following mean concerns or limitations:

1. If sEV were separated from bovine milk, and then used in the trail, results would be more convincible. As we know, milk contains more bioactive components other than sEV.

2. The conclusive idea, that miRs from milk are not transferred to circulation, is confuse. Reasons for this include: you have detected changes of ASC in piglets received bovine milk; miR29 and 22 were significantly increased in pig 3 6 hours after feeding milk(fig 9). Thus I suggest to change this conclusive idea.

By the way, the minor notices:

1.“CO2” is not correct.

2. I suggest to add more discussion on the absorption mechanism of digestive tract on milk sEV and its cargo miR or mRNA.

3. according to ISEV guideline, exosome should be replaced with sEV, small extracellular vesicle.

Reviewer #4: Review manuscript no. PONE-D-24-14376

Bovine Milk Consumption Affects the Transcriptome of Porcine Adipose Stem Cells: Do

Exosomes Play any Role?

Katherine Swanson, Jimmy Bell, David Hendrix, Duo Jiang, Brandon Batty, Melanie

Hanlon and Massimo Bionaz

General comment

The investigators studied the nutrigenomic effects of cow´s milk consumption on

the transcriptome of adipocyte stem cells (ASCs) of porcine subcutaneous adipose tissue after the piglet´s lactation period, with special attention to bovine milk exosomal microRNAs (miRs).

In detail: ASCs isolated from 19-20 week old piglets supplemented for 11 weeks with 750 mL of raw whole cow´s milk tended (P=0.10) to have a higher amount of ASCs isolated per gram of subcutaneous adipose tissue and a higher amount of CFU and ASC proliferation than control piglets fed an isocaloric protein-free maltodextrin solution.

Over 500 genes were differentially expressed (DEG) in ASCs isolated from bovine milk-fed vs. maltodextrin-fed control piglets. Bioinformatic analysis of DEG indicated an inhibition of the immune, neuronal, and endocrine systems and insulin-related pathways in ASCs of milk-fed piglets. Surprisingly, bioinformatic data point to an inhibition of the mTOR pathway in ASCs isolated from piglets consuming milk despite increased ASC proliferation. Experimental direct determination such as pS6K1 expression – as a measure of mTORC1 activity – have not been presented.

The investigators did not detect any exosomal miR or mRNA transfer from bovine milk to porcine-circulating plasma exosomes when investigating three 5-week-old Yorkshire Hampshire piglets implying that milk´s nutrigenomic effect on ASCs is not attributed to miRs in milk exosomes as observed after the lactation period.

The key message of the study is that cow´s milk consumption increased ASC numbers and ASC proliferation. However, the biological contribution of milk exosomal miRs on the epigenetic regulation of ASCs during lactation before and after weaning remains controversial and requires further investigations.

7. PLOS authors have the option to publish the peer review history of their article (what does this mean?). If published, this will include your full peer review and any attached files.

Reviewer #3: **Yes: **Yongliang Zhang

Reviewer #4: **Yes: **Bodo C. Melnik

---

## [Author Response · Author response to Decision Letter 1]

20 Nov 2024

The authors thank the reviewers for their time and insights. Please find our responses below. 

Review Comments to the Author

Reviewer #3: This paper present enough data, which may be interesting to readers. I think this ms had been well revised according to comments of reviewers. I deem it is acceptable, though it is of the following mean concerns or limitations:

1. If sEV were separated from bovine milk, and then used in the trail, results would be more convincible. As we know, milk contains more bioactive components other than sEV.

AU: The authors acknowledge that milk contains bioactive compounds that could also contribute to the results seen in this study. However, the overall objective of the study was to explore any differences in circulating miR and their effect on ASC transcriptome in pigs consuming milk compared with pigs consuming a maltodextrin solution. We have added one sentence in the conclusion to address this point.

2. The conclusive idea, that miRs from milk are not transferred to circulation, is confuse. Reasons for this include: you have detected changes of ASC in piglets received bovine milk; miR29 and 22 were significantly increased in pig 3 6 hours after feeding milk(fig 9). Thus I suggest to change this conclusive idea.

AU: Our data clearly shows no uptake of any miRNA in the blood of pigs. The change in miRNA in blood was not related to the milk miRNA. We do not know what affects those two miRNAs, but the effect was due to some other components in milk rather than miRNA vertically transferred from milk to circulation.

By the way, the minor notices:

1.“CO2” is not correct.

AU: Revised in manuscript

2. I suggest to add more discussion on the absorption mechanism of digestive tract on milk sEV and its cargo miR or mRNA.

AU: At this time, there is no known mechanism of absorption, just speculation

3. according to ISEV guideline, exosome should be replaced with sEV, small extracellular vesicle. https://www.isev.org/misev

AU: we checked on the isev web site but we could not find such a note. Rather, the isev indicated that exosomes are part of sEV.

Reviewer #4: Review manuscript no. PONE-D-24-14376

General comment

The investigators studied the nutrigenomic effects of cow´s milk consumption on

the transcriptome of adipocyte stem cells (ASCs) of porcine subcutaneous adipose tissue after the piglet´s lactation period, with special attention to bovine milk exosomal microRNAs (miRs).

In detail: ASCs isolated from 19-20 week old piglets supplemented for 11 weeks with 750 mL of raw whole cow´s milk tended (P=0.10) to have a higher amount of ASCs isolated per gram of subcutaneous adipose tissue and a higher amount of CFU and ASC proliferation than control piglets fed an isocaloric protein-free maltodextrin solution.

Over 500 genes were differentially expressed (DEG) in ASCs isolated from bovine milk-fed vs. maltodextrin-fed control piglets. Bioinformatic analysis of DEG indicated an inhibition of the immune, neuronal, and endocrine systems and insulin-related pathways in ASCs of milk-fed piglets. Surprisingly, bioinformatic data point to an inhibition of the mTOR pathway in ASCs isolated from piglets consuming milk despite increased ASC proliferation. Experimental direct determination such as pS6K1 expression – as a measure of mTORC1 activity – have not been presented.

The investigators did not detect any exosomal miR or mRNA transfer from bovine milk to porcine-circulating plasma exosomes when investigating three 5-week-old Yorkshire Hampshire piglets implying that milk´s nutrigenomic effect on ASCs is not attributed to miRs in milk exosomes as observed after the lactation period.

The key message of the study is that cow´s milk consumption increased ASC numbers and ASC proliferation. However, the biological contribution of milk exosomal miRs on the epigenetic regulation of ASCs during lactation before and after weaning remains controversial and requires further investigations.

AU: We thank the reviewer for thoroughly reviewing our manuscript and for the many suggestions and references provided!

Specific comments

Major flaws regarding the study design

The selected time period

To assess the impact of cow´s milk on adipocyte stem cell (ASC) proliferation in the porcine system, the physiology of porcine lactation and weaning has to be matched more closely. The nursing time of domestic pigs is 3-4 weeks, whereas wild pigs provide milk up to 16 weeks.

The study started with piglets 8- to 9 weeks of age exposed to cow´s milk for a further 11 weeks. Thus, the study may have missed a critical early window of postnatal milk-exosome-controlled ASC proliferation.

AU: For our study, we sought to use piglets as a model for young children, not infants. Therefore, we chose older weaned piglets that would consume milk supplementally, not as their primary form of nutrition.

Selected animals and postnatal time periods 

The reviewer wonders why the investigators used Duroc-Berkshire male piglets 8-9 weeks of age (n=6) for their milk challenge experiment but 5-week-old Yorkshire Hampshire piglets (n=3) for their exosome miR studies. Thus, the genetic background of the animals, the number of studied animals and timing of the sampling differ substantially, an avoidable mismatch for the later translation of obtained results.

Nevertheless, the use of 5 week old piglets may have compromised intestinal milk exosome and miR uptake and miR transmission into the circulatory system during this late stage of lactation (differences of colostrum milk exosomes versus mature milk exosomes). Differences in intestinal permeability and exosome composition and quantity have to be considered.

In 3–6 days old human neonates, intestinal permeability decreases in both term and preterm neonates.

van Elburg RM, Fetter WP, Bunkers CM, Heymans HS. Intestinal permeability in relation to birth weight and gestational and postnatal age. Arch Dis Child Fetal Neonatal Ed. 2003 Jan;88(1):F52-5. doi: 10.1136/fn.88.1.f52. PMID: 12496227; PMCID: PMC1755997.

Frazer LC, Good M. Intestinal epithelium in early life. Mucosal Immunol. 2022 Jun;15(6):1181-1187. doi: 10.1038/s41385-022-00579-8. Epub 2022 Nov 15. PMID: 36380094; PMCID: PMC10329854.

According to literature data, the intestinal closure in the piglet, i.e. the dramatic decrease in the transfer of macromolecules from the intestinal epithelium in the blood, is at about 24 h of age.

Ekström GM, Weström BR. Cathepsin B and D activities in intestinal mucosa during postnatal development in pigs. Relation to intestinal uptake and transmission of macromolecules. Biol Neonate. 1991;59(5):314-21. doi: 10.1159/000243365. PMID: 1714775.

Clarke RM, Hardy RN. Histological changes in the small intestine of the young pig and their relation to macromolecular uptake. J Anat. 1971 Jan;108(Pt 1):63-77. PMID: 5543214; PMCID: PMC1234227.

Ekström GM, Weström BR, Telemo E, Karlsson BW. The uptake of fluorescein-conjugated dextran 70,000 by the small intestinal epithelium of the young rat and pig in relation to macromolecular transmission into the blood. J Dev Physiol. 1988 Jun;10(3):227-33. PMID: 2464019.

Colostrum exosome transfer during the first hours and days of postnatal life may have already stabilized the intestinal permeability barrier limiting later transfer of mature milk exosomes.

Aarts J, Boleij A, Pieters BCH, Feitsma AL, van Neerven RJJ, Ten Klooster JP, M'Rabet L, Arntz OJ, Koenders MI, van de Loo FAJ. Flood Control: How Milk-Derived Extracellular Vesicles Can Help to Improve the Intestinal Barrier Function and Break the Gut-Joint Axis in Rheumatoid Arthritis. Front Immunol. 2021 Jul 28;12:703277. doi: 10.3389/fimmu.2021.703277. PMID: 34394100; PMCID: PMC8356634.

Tong L, Zhang S, Liu Q, Huang C, Hao H, Tan MS, Yu X, Lou CKL, Huang R, Zhang Z, Liu T, Gong P, Ng CH, Muthiah M, Pastorin G, Wacker MG, Chen X, Storm G, Lee CN, Zhang L, Yi H, Wang JW. Milk-derived extracellular vesicles protect intestinal barrier integrity in the gut-liver axis. Sci Adv. 2023 Apr 14;9(15):eade5041. doi: 10.1126/sciadv.ade5041. Epub 2023 Apr 12. PMID: 37043568; PMCID: PMC10096581.

Guo MM, Zhang K, Zhang JH. Human Breast Milk-Derived Exosomal miR-148a-3p Protects Against Necrotizing Enterocolitis by Regulating p53 and Sirtuin 1. Inflammation. 2022 Jun;45(3):1254-1268. doi: 10.1007/s10753-021-01618-5. Epub 2022 Jan 29. PMID: 35091894.

Melnik BC, Stremmel W, Weiskirchen R, John SM, Schmitz G. Exosome-Derived MicroRNAs of Human Milk and Their Effects on Infant Health and Development. Biomolecules. 2021 Jun 7;11(6):851. doi: 10.3390/biom11060851. PMID: 34200323; PMCID: PMC8228670.

Samuel M, Chisanga D, Liem M, Keerthikumar S, Anand S, Ang CS, Adda CG, Versteegen E, Jois M, Mathivanan S. Bovine milk-derived exosomes from colostrum are enriched with proteins implicated in immune response and growth. Sci Rep. 2017 Jul 19;7(1):5933. doi: 10.1038/s41598-017-06288-8. PMID: 28725021; PMCID: PMC5517456.

AU: We thank the reviewer for all the suggested references and comments! As indicated above and in our manuscript, we used piglets as models for human children. Thus, using different breeds should not be an issue but a strength, as humans are genetically diverse compared to pig breeds. If the vertical transfer of miRNA in children drinking milk happens, it should be evident by using an animal model with a very similar intestinal anatomy and function to humans. The time course experiment only confirmed what we observed with the main experiment; thus, it appears sufficient to allow us to conclude that if any vertical transfer of miRNA is happening, this would happen at a very low rate, at least less than 30% of the children consuming a very large amount of milk (i.e., did not happen in our 3 piglets). As discussed in our paper, we agree with the reviewer that the exosomes are likely vertically transferred during the nursing phase, while the intestinal permeability is present but becomes insignificant afterward. Our data confirmed such an idea, as argued in the discussion. Although we agree with the reviewer, our original hypothesis was that miRNAs are vertically transferred in children (or post-nursing piglets) and adults, which our data disproved. However, we have added most of the reviewer's suggestions and some of the suggested references to the revised discussion.

Weaning stress

It has recently been emphasized that weaning stress often causes changes in the morphology and function of the small intestine of piglets, disrupts digestion and absorption capacity, destroys intestinal barrier function, and ultimately leads to reduced feed intake, increased diarrhea rate, and growth retardation, a point that may be considered as well as the exosome study falls into this period.

Tang X, Xiong K, Fang R, Li M. Weaning stress and intestinal health of piglets: A review. Front Immunol. 2022 Nov 24;13:1042778. doi: 10.3389/fimmu.2022.1042778. PMID: 36505434; PMCID: PMC9730250.

At present, systematic studies investigating the time course of colostrum/milk exosome uptake from birth (time of colostrum exposure) to later periods of lactation (time of mature milk exposure) are missing. Individuals with inflammatory bowel diseases and leaky gut exhibit increased intestinal permeability, a constellation modifying milk exosome kinetics and uptake.

Michielan A, D'Incà R. Intestinal Permeability in Inflammatory Bowel Disease: Pathogenesis, Clinical Evaluation, and Therapy of Leaky Gut. Mediators Inflamm. 2015;2015:628157. doi: 10.1155/2015/628157. Epub 2015 Oct 25. PMID: 26582965; PMCID: PMC4637104.

AU: This is an interesting point to consider; however, all piglets in this study were healthy with no signs of gastrointestinal distress before being enrolled in our trial. There were minor instances of diarrhea when first introducing the milk and maltodextrin solution, but that was cleared up before the sampling period. This was discussed in our previous paper (see DOI: 10.3168/jds.2018-15201). We have added a comment to the discussion to address this point. 

Significant differences in protein intake during the intervention period 

The total protein content of the first porcine colostrum (16.65%) is approximately three times the level in porcine mature milk at the end of lactation (5.83%).

Csapó J, Martin TG, Csapó-Kiss ZS, Házas Z. Protein, fats, vitamin and mineral concentrations in porcine colostrum and milk from parturition to 60 days. Intern Dairy J 1966, 6 (8–9): 881-902; https://doi.org/10.1016/0958-6946(95)00072-0

After weaning pigs are physiologically not exposed to milk proteins and milk protein-derived signaling as well as milk exosomes and their potential miR-regulated gene expression.

During the intervention period of this study, piglets received 750 mL whole cow´s milk/day with 3,6% protein concentration resulting in 27 g milk protein uptake/day.

In comparison to the protein-free maltodextrin control, the milk-group received an additional total milk protein amounts of 2 kg providing substantial intake of essential branched-chain amino acids (BCAA) during the intervention period.

Only balancing the caloric intake between the milk group and the maltodextrin group is an oversimplified approach overlooking critical metabolic and epigenetic effects of BCAA-mTORC1 signaling.

BCAAs, enriched in milk proteins, are key activators of insulin and IGF-1 secretion as well as BCAA-mediated mTORC1 activation.

The authors provide no concrete data on mTORC1 activation such as the expression of the mTORC1 downstream target phosphorylated S6K1 (pS6K1). A Western blot analysis of pS6K1 should be provided to control the real impact of milk intake on mTORC1-S6K1 activation in ASCs.

High mTORC1-S6K1 signaling has been linked to increased adipogenesis by high protein formula feeding of human infants.

Melnik BC. Excessive Leucine-mTORC1-Signalling of Cow Milk-Based Infant Formula: The Missing Link to Understand Early Childhood Obesity. J Obes. 2012;2012:197653. doi: 10.1155/2012/197653. Epub 2012 Mar 19. PMID: 22523661; PMCID: PMC3317169.

Melnik BC. The potential mechanistic link between allergy and obesity development and infant formula feeding. Allergy Asthma Clin Immunol. 2014 Jul 22;10(1):37. doi: 10.1186/1710-1492-10-37. PMID: 25071855; PMCID: PMC4112849.

Exposure of infants to high protein formula is an accepted explanation of pediatrics for increased postnatal adipogenic programming by formula feeding.

Luque V, Closa-Monasterolo R, Escribano J, Ferré N. Early Programming by Protein Intake: The Effect of Protein on Adiposity Development and the Growth and Functionality of Vital Organs. Nutr Metab Insights. 2016 Mar 20;8(Suppl 1):49-56. doi: 10.4137/NMI.S29525. PMID: 27013888; PMCID: PMC4803318.

Haschke F, Grathwohl D, Detzel P, Steenhout P, Wagemans N, Erdmann P. Postnatal High Protein Intake Can Contribute to Accelerated Weight Gain of Infants and Increased Obesity Risk. Nestle Nutr Inst Workshop Ser. 2016;85:101-9. doi: 10.1159/000439492. Epub 2016 Apr 18. PMID: 27088337.

AU: We acknowledged in the manuscript that the increase in protein intake could play a role in the mTORC1 pathway. Our bioinformatic analysis of ASC actually found a slight inhibition of the mTORC1 pathway in pigs consuming milk. We have now further discussed in the manuscript these findings. Since the mTORC1 pathway was not significantly affected by diet in our study, we did not see a need to perform further analysis. Additional support for a lack of mTOR activation was the lack of effect on adiposity (see doi: 10.3168/jds.2018-15201). 

mTORC1 activation is required for differentiation of mesenchymal stem cells to ASCs

AU: We did not observe a significant change in the mTOR pathway due to diet in our study. 

Of note, activated mTORC1 plays a central role in the differentiation of mesenchymal stem cells to ASCs.

Xiang X, Zhao J, Xu G, Li Y, Zhang W. mTOR and the differentiation of mesenchymal stem cells. Acta Biochim Biophys Sin (Shanghai). 2011 Jul;43(7):501-10. doi: 10.1093/abbs/gmr041. Epub 2011 Jun 2. PMID: 21642276.

Zhou J, Li D, Wang F. Assessing the function of mTOR in human embryonic stem cells. Methods Mol Biol. 2012;821:361-72. doi: 10.1007/978-1-61779-430-8_23.

---

## [Decision Letter · Decision Letter 2]

3 Dec 2024

Bovine Milk Consumption Affects the Transcriptome of Porcine Adipose Stem Cells: do Exosomes Play any Role?

PONE-D-24-14376R2

Dear Dr. BIONAZ,

We’re pleased to inform you that your manuscript has been judged scientifically suitable for publication and will be formally accepted for publication once it meets all outstanding technical requirements.

Kind regards,

Wilfried A. Kues, Ph.D.

Academic Editor

PLOS ONE

Additional Editor Comments (optional):

Reviewers' comments:

Reviewer's Responses to Questions

**Comments to the Author**

1. If the authors have adequately addressed your comments raised in a previous round of review and you feel that this manuscript is now acceptable for publication, you may indicate that here to bypass the “Comments to the Author” section, enter your conflict of interest statement in the “Confidential to Editor” section, and submit your "Accept" recommendation.

Reviewer #3: All comments have been addressed

Reviewer #4: All comments have been addressed

2. Is the manuscript technically sound, and do the data support the conclusions?

Reviewer #3: Yes

Reviewer #4: Yes

3. Has the statistical analysis been performed appropriately and rigorously? 

Reviewer #3: Yes

Reviewer #4: Yes

4. Have the authors made all data underlying the findings in their manuscript fully available?

Reviewer #3: Yes

Reviewer #4: Yes

5. Is the manuscript presented in an intelligible fashion and written in standard English?

Reviewer #3: Yes

Reviewer #4: Yes

6. Review Comments to the Author

Reviewer #3: I think authors have tried best to revise this ms. Since this paper present a negative retults compared to the idea that cow milk sEV may regulate recipient via miRNA, it may be helpful in further understanding how cow milk sEV functions in recipeints. Thus, I suggest it be accepted.

Reviewer #4: The revised version of the manuscript improved substantially. The authors answered all major issues raised by the reviewers and provided an updated reference list of the literature. The reviewer has no further comments.

7. PLOS authors have the option to publish the peer review history of their article (what does this mean?). If published, this will include your full peer review and any attached files.

Reviewer #3: No

Reviewer #4: No

---

## [Editor Report · Acceptance letter]

10 Dec 2024

PONE-D-24-14376R2 

PLOS ONE

Dear Dr. Bionaz, 

I'm pleased to inform you that your manuscript has been deemed suitable for publication in PLOS ONE. Congratulations! Your manuscript is now being handed over to our production team.

Kind regards, 

on behalf of

Dr. Wilfried A. Kues 

Academic Editor

PLOS ONE